# Digital health solutions and integrated COVID-19 and TB services to help recover TB care and prevention services in the COVID-19 pandemic: A descriptive study in four high TB burden countries

**Ineke Spruijt**[1]*, **Yalda Alam**[1], **Huong Nguyen**[2], **Bakyt Myrzaliev**[3], **Muratbek Ahmatov**[3], **Bethrand Odume**[4], **Lillian Mtei**[5], **Agnes Gebhard**[1], **Mustapha Gidado**[1], **Degu Jerene**[1]

1 Technical Division, KNCV Tuberculosis Foundation, The Hague, The Netherlands, 2 Technical Division, KNCV Vietnam, Hanoi, Vietnam, 3 Technical Division, KNCV Kyrgyzstan, Bishkek, Kyrgyzstan, 4 Technical Division, KNCV Nigeria, Abuja, Nigeria, 5 Technical Division, KNCV Tanzania, Dar es Salaam, Tanzania

* ineke.spruijt@kncvtbc.org

## Abstract

### Background

The measures undertaken to control COVID-19 have disrupted many platforms including tuberculosis (TB) healthcare services. Consequently, declines in TB notifications have been observed in various countries. We visualized changes over time in TB and SARS-CoV-2 infection notifications and reported on country-specific strategies to retain TB care and prevention services in Kyrgyzstan, Nigeria, Tanzania, and Vietnam.

### Methods

We collected and visualized quarterly, retrospective, and country-specific data (Quarter (Q) 1 2018- Q1 2021) on SARS-CoV-2 infection and TB notifications. Additionally, we conducted a country-specific landscape assessment on COVID-19 measures, including lockdowns, operational level strategy of TB care and prevention services, and strategies employed to recover and retain those services. We used negative binomial regression models to assess the association between the installation of COVID-19 measures and changes in TB notifications.

### Results

TB notifications declined in Kyrgyzstan and Vietnam, and (slightly) increased in Nigeria and Tanzania. The changes in TB notifications were associated with the installation of various COVID-19 prevention measures for Kyrgyzstan and Vietnam (declines) and Nigeria (increases). All countries reported reduced TB screening and testing activities. Countries reported the following strategies to retain TB prevention and care services: digital solutions for treatment adherence support, capacity building, and monitor & evaluation activities;

**Data Availability Statement:** All relevant data are within the manuscript and its Supporting Information files.

**Funding:** IS, DJ, AG, and MG received funding from the 's Gravenhaagsche association to conduct this study. The funder had no role in study design, data collection and analysis, decision to publish or preparation of the manuscript.

**Competing interests:** The authors have declared that no competing interests exist.

adjustment in medication supply/delivery & quantity, including home delivery, pick up points, and month supply; integrated TB/COVID-19 screening & diagnostic platform; and the use of community health care workers.

## Conclusion

Following the COVID-19 pandemic, we did not observe consistent changes in TB notifications across countries. However, all countries reported lower operating levels of TB prevention and care services. Digital health solutions, community-based interventions, and the integration of COVID-19 and TB testing services were employed to recover and retain those services.

## Introduction

The sudden emergence of the novel SARS-Cov-22-virus, causing corona virus disease 2019 (COVID-19) has negatively impacted the fight against tuberculosis (TB). Countries have reported considerable declines in TB notification rates: in 2021, the global TB notification rate declined by 18% and is now back at 2012 level [1]. One factor causing this decline is the disruption of TB services: 42% of 105 surveyed countries reported a variety of disruptions in their TB care and prevention services related to strict regulations and imposed COVID-19 infection and prevention control (IPC) measures [2]. Mathematical models predicted that disruptions of TB care and prevention services could lead to an increase in TB mortality of 5–15% over the next 5 years [3]. The COVID-19-related disruptions in healthcare systems are reversing global progress made in the fight against TB [4]. Consequently, actions to reverse these impacts from COVID-19 and retain TB prevention and care activities are urgently needed [1].

It can be argued that the COVID-19 IPC measures (e.g., social distancing; improved hygiene; mandatory masking; and lockdown) also limit the transmission of other pathogens causing respiratory infectious disease, such as TB. Consequently, reduced TB transmission could also contribute to a decline in TB notifications. However, this is unlikely as it is estimated that 5% of people with primary TB infections will develop TB disease within the first 1.5 years, and 5% will do so throughout their remaining lifetime [1, 5, 6]. Therefore, effects on TB notification trends are not expected in such short period of time. Thereby, any effect from reduced transmission due to social distancing on TB deaths is estimated to be outweighed by the adverse effects of disrupted health systems [3].

Since both COVID-19 and TB are respiratory infectious diseases with overlapping symptoms (cough, fever, night sweats, loss of appetite), there are opportunities to recover and retain TB prevention and care services by integrating COVID-19 and TB IPC measures and testing practices [7, 8]. Additionally, engagement of communities, youth, and civil society could aid in closing testing and care gaps for both COVID-19 and TB services [7, 8]. Assessing countries' experiences and identifying lessons learned from integrating these activities could inform decisions and stimulate other countries to adopt similar approaches. Therefore, during the period of 2018–2021 we visualized and evaluated changes in national TB notifications and practical operations of TB prevention and care services (including the initiation of TB and COVID-19 integrated services) in Kyrgyzstan, Nigeria, Tanzania, and Vietnam.

## Materials and methods

### Study design and setting

We performed a cross sectional descriptive study to evaluate changes related to the installment of COVID-19 IPC measures in national TB notifications and practical operations of TB prevention and care services (including the initiation of TB and COVID-19 integrated services)in four high TB burden countries: Kyrgyzstan, Nigeria, Tanzania, Vietnam [1]. These countries were selected purposively to ensure geographic representation across high TB burden settings. KNCV Tuberculosis Foundation is a non-governmental organization with headquarters in The Netherlands, and country offices in (amongst others) Kyrgyzstan, Nigeria, Tanzania, and Vietnam. KNCV's country presence was taken into consideration during selection of the study countries to ensure high quality data collection.

### Data collection

**Landscape assessment.** We asked project members -KNCV country office representatives–to conduct a landscape assessment for their countries to assess 1) the quarterly installment of nationwide COVID-19 IPC measures (Tables 1 and 2) the overall changes in operations of TB care and preventions services and facilities and 3) the integration of COVID-19 and TB activities during Quarter (Q)1 2020-Q4 2021 period. COVID-19 IPC measures consisted of 1) any type of lockdown; 2) social distancing ($\geq$1 meter); 3) mandatory masking in public spaces; 4) limited availability of public transportation (i.e., reduced availability of public transportation compared to the pre-COVID pandemic situation). Table 1 provides a country-specific definition of lockdown and social distancing. The landscape assessment consisted of an online questionnaire filled in by the country representatives which we designed in Microsoft Forms. S1 Questionnaire provides an overview of the questionnaire filled in as part of the landscape assessment.

**Country-specific TB notification data.** Each quarter, the World Health Organization (WHO) receives data on the number of newly diagnosed TB patients (TB notification data) from country's National TB Programs (or equivalents), who receive those data from relevant health authorities [9]. We collected quarterly TB notification data for the period Q1 2018 until Q1 2021. First, we collected data publicly available data from the Global Tuberculosis Programme (e.g., period Q1 2020 until Q4 2021 for all countries). Second, as quarterly data was not publicly available through the WHO's Global TB Programme for the period prior to Q1 2020, we designed a data entry tool in Microsoft Excel and approached KNCV's country representatives to collect those data from the NTPs.

*Country-specific SARS-Cov-2 infection notification data.* We collected publicly-available data on daily notified SARS-Cov-2 infections and calculated quarterly data for the period Q1 2020 until Q4 2021 from the database "Our World in Data" on the "Coronavirus Pandemic (COVID-19)" [10].

### Data analyses

First, we used Microsoft Excel to visualize and evaluate country-specific changes in SARS-CoV-2 infection and TB notifications during the period Q1 2018 and Q4 2020. Second, we described outcomes of the landscape assessments on TB and COVID-19 integrated responses and services. Third, we used a univariable negative binomial regression model to assess the associations between the quarterly installation of one or more COVID-19 IPC measures (independent variables; categorical data) and the quarterly changes in TB notifications (dependent variable; count data). We included the following COVID-19 IPC measures in the model:

**Table 1. Country-specific definitions of COVID-19 IPC measures.**

| **Kyrgyzstan** | |
|---|---|
| Lockdown | It is forbidden to: <br> · To visit relatives, acquaintances, neighbors, and colleagues; <br> · Gather in groups of more than three people; <br> · To move around the city for minors unaccompanied by adults <br> · Suspension of the work of all organizations and enterprises, with the exception of organizations and enterprises that ensure the vital activity of the city/district; <br> · Limit the operation of entertainment establishments (nightclubs, restaurants, catering outlets, cafes, karaoke, bars), including those located in shopping and entertainment centers, from 23:00–07:00 without changing the mode of providing delivery services food and scale of takeaway food until the epidemiological situation improves. <br> · No movement of people and personal vehicles unless absolutely necessary. |
| Social distancing | Keeping distance 1–1.5 meters between people, and avoid close contact including handshakes, hugs, and kisses. |
| **Nigeria** | |
| Lockdown | people are expected to stay at home. Also, location of contacts such as schools, universities, hotels, clubs, and religious houses are closed, social gatherings involving above 20 people are prohibited and economic activities involving physical interaction are halted and ban on international flights |
| Social distancing | A strategy to reduce physical contact between people with the aim of slowing down and reducing the spread of COVID-19 in a community. It involves strict adherence to; <br> a. Non-physical greetings (avoiding hand shaking or hugs) <br> b. Maintaining at least 2 meters (6 feet) physical distance between yourself and individuals and; <br> c. Closure of activities that will cause any form of gathering including schools, places of worship, sporting and social events. |
| **Tanzania** | |
| Lockdown | A government regulation restricting movement and normal day to day activities of people. In Tanzania the government never gave restrictions for a lockdown throughout the pandemic. People were allowed to continue with all activities without any hindrance. [Interestingly, the public itself decided to restrict their own movements because of fear; in Q2 and Q3 2020 most people literally locked themselves up.] |
| Social distancing | A government regulation that when in public there should be a distance between people of at least 1m. This was never put forward by the government, but a lot of institutions (public and private) decided themselves to mandate people to sit/stand in distances of 1m or more. The government did not enforce it though. |
| **Vietnam[1]** | |
| Lockdown | Directive 16. Social isolation–everyone must stay at home, only going out when absolutely necessary. Do not gather more than 2 people outside of offices, hospitals, schools and public places. <br> Directive 15. Stop gathering at events with more than 20 people per room. Stop all cultural activities, sports, entertainment activities, Do not gather with >10 people outside of offices, hospitals, and schools. |
| Social distancing | Directive 15 & 16: at least 2 meters distance |

[1] Directives of the Prime Minister on measures to prevent and control the Covid-19 epidemic such as mass gathering, minimum safe distance, operation of business establishments, transportation

lockdown (yes/no); mandatory masking in public (yes/no); social distancing ($\geq$1 meter distance) (yes/no); limited availability of public transport (yes/no). Statistical analysis was carried out for Kyrgyzstan, Nigeria, and Vietnam. We did not perform analysis for Tanzania due to the lack of installment of COVID-19 IPC measures. During exploratory analysis, we detected overdispersion in the notification data: the variance of the response variable (TB notifications) exceeded that of its mean. To account for this overdispersion, we employed a negative binomial regression model with log-link functions to estimate incidence rate ratios (IRR) with 95% confidence intervals [11]. For Kyrgyzstan (the only country with multiple significant IPC measures in univariable analysis), we fitted a multivariable model using backward elimination of

**Table 2. Operational level strategy of tuberculosis diagnostic and treatment services during the pandemic and compared to the pre-COVID-19 pandemic situation per country.**

| Type of service | Operational level strategy[1] compared to the pre-COVID-19 pandemic situation per country | | | |
|---|---|---|---|---|
| | Kyrgyzstan | Nigeria | Tanzania | Vietnam |
| Active TB case finding / screening | Lower | Lower | Lower | Lower |
| DS-TB diagnostic services | Lower | Same | Same | Same |
| DR-TB diagnostic services | Lower | Same | Same | Same |
| DS-TB treatment services | Lower | Same | Same | Same |
| DR-TB treatment services | Lower | Same | Same | Same |
| TB preventive services/treatment | Lower | Same | Same | Same |

TB: tuberculosis; DS-TB: drug-susceptible tuberculosis; DR-TB: drug-resistant tuberculosis

Operational level strategy: the means used by organizations and services to accomplish their overall objectives.

the initial model with variables yielding a p-value <0.2 in univariable analysis, guided by changes in regression coefficients and in the model fit. We used SPSS Statistics 26.0 (IBM Analytics, Chicago, IL, USA) for data management and analysis. S1 File provides a country profile including detailed results of the changes in SARS-CoV-2 and TB notification data and the landscape assessment.

## Ethics

This study included anonymized aggregated surveillance data which were provided by the country-specific National TB Programs to project members and KNCV country representatives or collected through publicly available datasets. Furthermore, project members who served as KNCV country representatives conducted a landscape assessment in which they reported on past or present COVID-19 restrictions and prevention measures, which are publicly known. Since this study did not involve human subjects and/or medical interventions the declaration of Helsinki is not applicable to this study and therefore we did not obtain informed consent from KNCV country representatives for the execution of the landscape assessment, nor did we seek approval from an Ethics-committee.

## Results

### Descriptive analysis

**SARS-CoV-2 infection notifications and prevention measures.** Following the emergence of the COVID-19 pandemic in Q1 2020, the number of notified SARS-CoV-2 infections increased most rapidly from 5399 to 41163 infections (662%)) between Q2 2020-Q3 2020 in Kyrgyzstan; from 28759 to 75284 infections (162%) between Q4 2020-Q1 2021 in Nigeria; and from 14449 to 773703 infections (5255% increase) between Q2 2021 and Q3 2021) in Vietnam. The increase in SARS-CoV-2 infections in Tanzania started after Q2 2021 (from 0 to 25337 infections) Vietnam (Fig 1). Kyrgyzstan, Nigeria and Vietnam, had COVID-19 IPC measures in place during the period Q1 2020 and Q1 2021. S1 File provides detailed country-profiles.

**TB notifications.** The number of TB notifications declined during the period Q1 2020 and Q3 2020 for Kyrgyzstan from 1839 to 915 (50% decline) and for Vietnam from 23073 to 9273 (60% decline) (Fig 1). During the period Q1 2020 and Q3 2021, Tanzania showed fluctuating TB notifications and Nigeria showed an overall increase in TB notifications (from 10927 in Q1 2020 to 13269 in Q1 2021 (increase of 21%)) (Fig 1). TB treatment initiation and completion data were available for Kyrgyzstan, Nigeria, and Tanzania and showed a trend

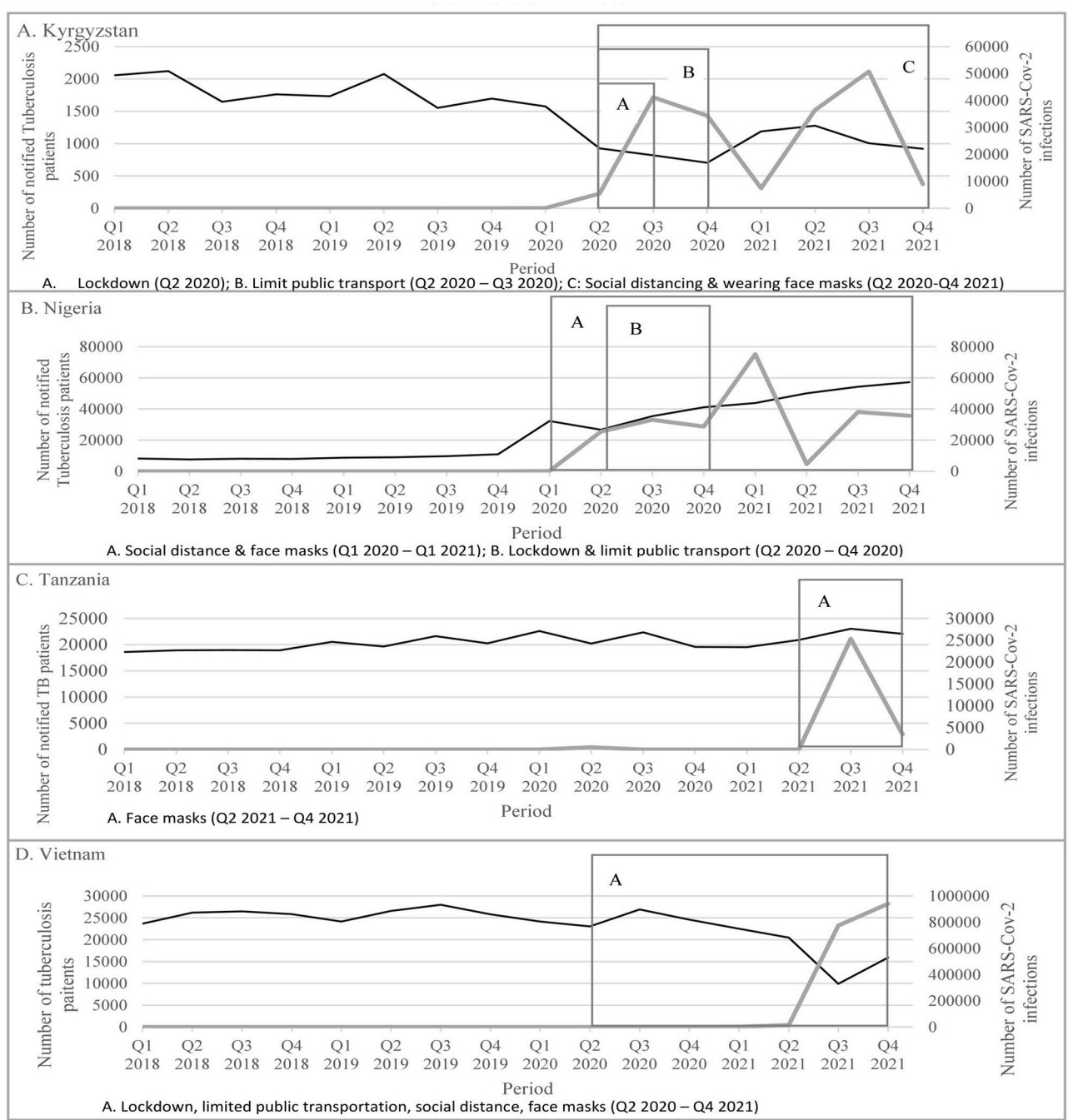

**Fig 1. Changes over time in TB and SARS-CoV-2 infections notifications, including COVID-19 prevention measures during the Q1 2018 and Q1 2021 period.**

comparable to the TB notification trend: declining for Kyrgyzstan, slightly increasing for Nigeria, and fluctuating for Tanzania. (S1 File)

**TB prevention and care services.** For Kyrgyzstan, all TB diagnostic and treatment services were reported to operate at a lower operational level (e.g., using less means to accomplish objectives) when compared to pre-pandemic operational levels (Table 2). For Nigeria, Tanzania, and Vietnam, only active TB case finding- and screening services were reported to operate

**Table 3. Experienced shortages during COVID-19 pandemic (Quarter 1 2020-Quarter 4 2021).**

| Shortages | Country | | | |
|---|---|---|---|---|
| | **Kyrgyzstan** | **Nigeria** | **Tanzania** | **Vietnam** |
| TB care staff (redeployed to COVID-19 activities) | Yes | Yes | No | Yes |
| PPE materials | No | No | No | No |
| TB diagnostics | Yes | No | No | No |
| DS-TB medication | No | Yes | No | No |
| DR-TB medication | No | Yes | No | No |

PPE: personal protective equipment; TB: tuberculosis; DS-TB: drug-susceptible tuberculosis; DR-TB: drug-resistant tuberculosis

at a lower operational level, whereas other services remained at similar levels when compared to pre-pandemic levels. Additionally, shortages of TB diagnostics (Kyrgyzstan) and Drug-Susceptible (DS)-TB and/or Drug-Resistant (DR)-TB medication (Nigeria) were reported (Table 3). Furthermore, human resources in TB prevention and care services were also reported to be affected by the COVID-19 pandemic in three countries (Kyrgyzstan, Nigeria, Vietnam). Reported reasons for this are: 1) TB-healthcare worker (HCW) refrained from work out of fear for COVID-19 (Nigeria); 2) TB-HCW were redeployed to COVID-19 activities (Kyrgyzstan, Nigeria, Vietnam); 3) increased sick leave of TB-HCW (Kyrgyzstan); 4) inability to attend work due to lockdown/restricted public transportation (Nigeria). Furthermore, Nigeria deployed lay providers and non-healthcare workers to assist in TB prevention and care activities and encouraged the continuity of TB care and prevention activities by providing personal protective equipment (PPE) to healthcare facilities.

**Strategies to recover or retain TB prevention and care services.** *Digital solutions.* Digital health tools were employed to retain monitoring and evaluation activities and TB diagnostic and treatment services. For example, diagnostic outcomes were communicated by HCWs to patients through SMS (Nigeria, Tanzania, Vietnam), video observed treatment (VOT) was increasingly used for treatment services in Kyrgyzstan, and treatment support was provided through telephone calls (Nigeria, Tanzania, Vietnam). Furthermore, digital strategies were also used for capacity building. For example, HCWs, community-workers / lay-providers were virtually trained on integrating COVID-19 and TB responses and activities and providing treatment support (Kyrgyzstan, Nigeria, Tanzania, Vietnam) (Table 4).

**Remote TB care and prevention services.** All countries reported increased use of community-workers/volunteers/non-healthcare workers to provide community-based treatment support. In Tanzania, facilities operated home collection and transportation services for diagnostic specimens. Furthermore, healthcare facilities in Nigeria limited the need for patients to travel to pharmacies by making more use of community-based treatment monitor and medication refill services.

**Integrated TB/COVID-19 services.** In Nigeria and Vietnam, COVID-19 and TB responses were integrated by adopting an integrated COVID-19/TB screening for "emergency use approval". The screening approach, which consists of TB screening using chest X-ray and confirmatory Gene Xpert testing and COVID-19 screening using antigen testing, is currently under consideration to be included in the national GeneXpert guideline in Nigeria. In Vietnam, TB screening was integrated into COVID-19 vaccination sites under the USAID Erase TB project [12]. Furthermore, healthcare facilities and/or community health posts in Nigeria and Vietnam organized combined COVID-19 and TB screening events to increase or retain diagnostic capacity. Finally, healthcare facilities in all countries were reported to have adopted integrated COVID-19 and TB IPC measures to continue services.

**Table 4. Strategies implemented to retain TB services.**

| Strategies implemented to retain services | Country | | | |
|---|---|---|---|---|
| | **Kyrgyzstan** | **Nigeria** | **Tanzania** | **Vietnam** |
| **IPC measures employed to retain TB services** | | | | |
| Integrated COVID-19 and TB IPC | Yes | Yes | Yes | Yes |
| Providing PPE to healthcare facilities | No | Yes | No | No |
| **Screening and diagnostic services** | | | | |
| • COVID-19 and TB integrated screening and diagnostic capacity | No | Yes[1] | No | Yes |
| **Digital solutions** | | | | |
| • Communication of diagnostic outcomes through SMS | No | Yes | Yes | Yes |
| • Increased use of Video observed treatment (VOT) | Yes | No | No | Yes |
| • Telephone follow-up treatment support | No | Yes | Yes | Yes |
| • Digital M&E meetings | Yes | Yes | No | No |
| • Increased real time TB notification and treatment outcome data | No | No | Yes | No |
| • Digital notification tools | No | Yes | No | No |
| • Virtual training of healthcare workers / community leaders in COVID-19 and TB integrated responses and activities | Yes | Yes | No | Yes |
| **Remote TB care and prevention services** | | | | |
| • Home collection and transportation of specimen | No | No | Yes | No |
| • Treatment support through community-workers/volunteers/non-healthcare workers | Yes | Yes | Yes | Yes |
| • Multi-month medication dispensing | No | No | No | No |
| • Medication refill and monitoring services | No | Yes | No | No |

IPC: infection prevention and control; M&E: monitoring & evaluation; PPE: personal protective equipment; TB: tuberculosis

[1] During the COVID-19 pandemic, the Presidential Task Force (PTF), Nigerian Center for Disease Control and the National TB Program developed and adopted "integrated COVID-19/TB screening approach" as Emergency Use Approval. The National TB Program is currently revising the existing National GeneXpert guideline to include this integrated COVID-19/TB screening approach.

### The association between COVID-19 IPC measures and TB notifications

For Kyrgyzstan, the installation of combined policies for mandatory masking in public and social distancing ($\geq$1 meter) was significantly associated in multivariable analysis with declines in the number of TB notifications those quarters. For Vietnam, the installation of combined policies (e.g., lockdown, mandatory masking, social distancing ($\geq$ 1 meter), and limited availability of transportation) was associated with declines in the number of TB notifications during the period Q2 2020 until Q4 2021. Contrary, for Nigeria, the installation of combined policies for mandatory masking in public and social distancing ($\geq$1 meter) was significantly associated with increases in the number of TB notifications (Table 5).

### Discussion

We retrospectively visualized and evaluated changes over time in SARS-CoV-2 infections and TB notifications and reported retainment strategies for TB services and COVID-19 and TB integrated approaches during the period 2018–2021 in four high TB burden countries. We observed a decline in TB notifications for Kyrgyzstan and Vietnam. We did not find a consistent association between the installation of COVID-19 IPC measures and a change in TB notifications: for Kyrgyzstan the installation of various COVID-19 IPC measures was associated with a decreasing TB notification, whereas for Nigeria this was associated with increasing TB notifications. Since the emergence of the COVID-19 pandemic, all countries reported a lower operational level of TB care and prevention services. To retain these services, all countries

**Table 5. Associations between COVID-19 measures and notification of TB patients.**

| COVID-19 prevention measure by country | | | Negative Binomial Regression Model (log link) |
|---|---|---|---|
| | | | crude IRR (95%CI) |
| **Kyrgyzstan** | **Lockdown (Q2 2020)** | No | REF |
| | | Yes | 0.69 (0.40–1.21) |
| | **1) Mandatory masking in public and 2) social distance (≥1 meter) (Q2 2020-Q4 2021)** | No | REF |
| | | Yes | 0.58 (0.52–0.65)[1] |
| | **Limited availability of public transportation (Q2 2020-Q3 2020)** | No | REF |
| | | Yes | 0.63 (0.44–0.91) |
| **Nigeria** | **1) Lockdown and 2) limited availability of public transportation (Q2 2020-Q3 2020)** | No | REF |
| | | Yes | 1.25 (0.42–3.74) |
| | **1) Mandatory masking in public and 2) social distance (≥1 meter) (Q1 2020-Q4 2021)** | No | REF |
| | | Yes | 4.85 (4.02–5.84) |
| **Vietnam** | **1) Lockdown, 2) limited availability of public transportation, 3) Mandatory masking in public and 4) social distance (≥1 meter) (Q2 2020-Q4 2021)** | No | REF |
| | | Yes | 0.79 (0.65–0.96) |

CI: confidence interval; IRR: incidence rate ratio; REF: reference category

[1] Only variable that remained significant in multivariable analysis

reported the employment of remote or digital treatment support activities and COVID-19 and TB integrated responses (except for Tanzania).

Similar to a previous WHO report, we showed a decline in TB notification for two of four countries [1, 7, 8]. In line with other studies, the decline in Kyrgyzstan was associated with the imposition of COVID-19 IPC measures. For example, during the South African lockdown declines were observed in weekly notified and microbiologically confirmed TB patients and the number of TB tests [13]. Similarly in India and Malawi, TB notifications declined following the installation of COVID-19 IPC measures [4, 14]. Another possible explanation for reduced TB notification is a change in healthcare seeking behaviors. A review from Dlangala et al. (2021) showed that both patients and healthcare workers struggled to access primary care (and a consequent shift of care seeking to informal care facilities) and feared for contracting COVID-19 infection [15]. Furthermore, we reported disrupted TB care and preventions services (particularly TB screening and testing) services. This is in line with a global WHO survey, which reported that 90% of countries reported a disruption in the health system with greatest disruptions in low-and-middle income countries [2].

Our study showed increases in TB notifications for Nigeria. This is not in line with a study by Odume et al. (2020), who evaluated two active case finding interventions and showed a decrease in clinic attendance, presumptive TB identification, TB cases detected and TB treatment initiation during the first quarter of the COVID-19 pandemic [16]. These findings were supported by Adwole (2020), who also showed reductions of 35% in presumptive TB and 34% in active TB cases detected in a health facility during the same period. News reports confirm the increase in TB notifications, which were attributed to programmatic and facility-based interventions and integration of TB and COVID-19 measures [17]. Lacking scientific evidence, it is unclear to what extend new and additional efforts may have outweighed or masked the effects of COVID-19 on the TB epidemiology in Nigeria.

The WHO has released five key actions to be undertaken to contain the continuity of TB prevention and care services: 1) ensure effective IPC measures to protect HCWs and patients; 2) scale-up of integrated COVID-19 and TB testing; 2) promote community-based and home-based people-centered prevention and care services; 3) stand against stigma and promote

human rights; 4) build and strengthen community, youth and civil society engagement to close gaps in care [7, 8]. Our study showed that all four countries have followed this advice and have employed multiple strategies to recover and retain TB prevention and care services, such as the provision of PPE to TB services; engagement of community-leaders; remote and home-based digital treatment support; and the training of community-workers and lay-workers. Furthermore, the training and engagement of community-leaders and workers could also address stigma. COVID-19 and TB stigma has already been linked to each other, emphasizing the importance for integrated interventions to address the stigma. Furthermore, following some similarities in the response to COVID-19 and TB, TB prevention and care activities can be well integrated and/or employed with COVID-19 responses [18]. Our study showed that three out of four countries integrated COVID-19 and TB screening services and training on those activities.

Our study was only able to statistically assess the association between COVID-19 restrictions and changes in TB notifications in two out of four countries. Additionally, only SARS-Cov-2 diagnosed infections are reported, which may be an underestimation of the real burden of COVID-19. This may be particularly the case in countries where there is limited SARS-Cov-2 testing facilities available. Furthermore, we could not assess the magnitude of factors such as decline in care seeking behavior; shortages in TB care and prevention supplies; and disruption in TB services; nor could we estimate the impact of those factors on TB notification. To assess the impact of COVID-19 IPC measures on TB notifications, future research is needed that take into account those factors and their magnitudes. Furthermore, future research should explore and take into account any programmatic and project interventions aimed at increasing TB notifications that were implemented simultaneously with the COVID-19 IPC measures. Although this study could not assess the impact of COVID-19 IPC measures on TB notifications, this descriptive study does give important insights in the strategies that countries employed to retain their TB care and prevention services and the opportunities to integrate COVID-19 and TB services. We recommend future mixed methods studies to study best practices of integrating COVID-19 and TB services and their effects on TB and SARS-CoV-2 infections notification and treatment. Furthermore, we recommend future studies to consider mix-method with a qualitative section addressing fear and stigma during pandemics or outbreaks and the effects on healthcare seeking behavior.

## Conclusion

TB notifications decreased in two of four countries, which could be caused by reported disruption in TB care and prevention services -especially TB screening and testing services and shortages in supplies. Multiple digital health solutions, community-based interventions (such as community-workers for treatment support and home-delivery services,) and the integration of COVID-19 and TB testing services have great potential to recover and retain TB care and prevention services.

## Supporting information

**S1 Questionnaire. Questionnaire landscape assessment.**
(PDF)

**S1 File. Country profiles.**
(PDF)

## Acknowledgments

We would like to thank the National TB Program managers of Nigeria, Tanzania, and Kyrgyzstan for providing quarterly TB notification and treatment data.

## Author Contributions

**Conceptualization:** Ineke Spruijt, Degu Jerene.

**Data curation:** Ineke Spruijt, Yalda Alam, Huong Nguyen, Bakyt Myrzaliev, Muratbek Ahmatov, Bethrand Odume, Lillian Mtei.

**Formal analysis:** Ineke Spruijt, Yalda Alam.

**Funding acquisition:** Agnes Gebhard, Mustapha Gidado, Degu Jerene.

**Investigation:** Ineke Spruijt.

**Methodology:** Ineke Spruijt.

**Supervision:** Ineke Spruijt, Degu Jerene.

**Writing – original draft:** Ineke Spruijt.

**Writing – review & editing:** Ineke Spruijt, Huong Nguyen, Bakyt Myrzaliev, Muratbek Ahmatov, Bethrand Odume, Lillian Mtei, Agnes Gebhard, Mustapha Gidado, Degu Jerene.

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
