## [Decision Letter · Decision Letter 0]

20 Jul 2022

PONE-D-22-12851Digital health solutions and integrated COVID-19 and TB services to help recover TB care and prevention services in the COVID-19 pandemic: a descriptive study in four low-middle-income countriesPLOS ONE

Dear Dr. Spruijt,

Thank you for submitting your manuscript to PLOS ONE. After careful consideration, we feel that it has merit but does not fully meet PLOS ONE’s publication criteria as it currently stands. Therefore, we invite you to submit a revised version of the manuscript that addresses the points raised during the review process.

 Thank you to the authors for this contribution for consideration, the content of which is interesting and topical in light of post pandemic recovery for all public health issues, including tuberculosis. That said, some major issues of concern have been raised by reviewers, in particular with regards to editing, consistency, and clarity. Some questions have been raised as to the methodology, and inclusion of superfluous data. I urge specific attention to these issues in a major revision. 

We look forward to receiving your revised manuscript.

Kind regards,

Courtney Elizabeth Heffernan

Academic Editor

PLOS ONE

Journal Requirements:

Reviewers' comments:

Reviewer's Responses to Questions

**Comments to the Author**

1. Is the manuscript technically sound, and do the data support the conclusions?

Reviewer #1: Partly

Reviewer #2: Partly

Reviewer #3: Partly

2. Has the statistical analysis been performed appropriately and rigorously? 

Reviewer #1: No

Reviewer #2: No

Reviewer #3: Yes

3. Have the authors made all data underlying the findings in their manuscript fully available?

Reviewer #1: No

Reviewer #2: Yes

Reviewer #3: Yes

4. Is the manuscript presented in an intelligible fashion and written in standard English?

Reviewer #1: Yes

Reviewer #2: Yes

Reviewer #3: No

5. Review Comments to the Author

Reviewer #1: I have read with great interest the study by Spruijt and colleagues regarding the impact of COVID-19 on TB services in four high TB burden countries. The research question is of utmost important and the study provides timely findings, showing the importance of implementing digital tools to counteract the negative consequences of the pandemic on TB prevention and care. The article is well-written but would benefit from revision, with particular focus on the methods. In fact, I believe that a more detailed description of the approaches used to address the study objectives would help interpretation and enable replication/reproducibility. Most of the study is qualitative in nature, aside from the use of statistical modelling to determine associations between various COVID-19-related measures and TB notification rates. The dual nature of the study, which allowed to provide insights into the reasons behind certain events and measures taken to retain TB activities in spite of the ongoing challenges, could be better emphasized as a strength of the study.

Please see more detailed comments/questions below:

Major comments:

1) Abstract: Only three countries are mentioned in the background, but a fourth country (Vietnam) appears in the results, which generates confusion. Also, I think it would be nice to briefly indicate the data sources in the methods, so that readers understand right away whether the study being reported pertains to the national context or specific areas within each country.

2) Methods, lines 106-108: I wondered whether the authors could elaborate a bit more on the methods used to conduct the landscape assessment. Were TB representatives interviewed or invited to fill out a written questionnaire?

3) Methods, lines 124-125: I am not convinced that using the term “trends” is appropriate here, given that – to my understanding – no statistical analyses were undertaken to examine trends (i.e., trend tests or other approaches). I would lean towards using a less specific terminology, e.g., “change over time”.

4) Methods, lines 128-130: I think that adding a brief explanation about the reasons for choosing the Eta coefficient would be helpful to the reader (perhaps the non-linear nature of the relationship between variables?). In addition, “correlation” and “association” are not synonyms, so I would not talk about “association” here as the Eta coefficient only reflects correlation.

5) Methods, line 133: While I do understand the need for a negative-binomial model instead of Poisson given the problem of overdispersion, I think the model’s description should be expanded a bit to clarify 1) the purpose of this analysis, 2) the model specification (i.e. which variables were included and why/how these were selected), the estimated measures (crude/adjusted incidence rate ratios and 95% CIs?).

6) It remains unclear to me whether and how data on COVID-19 control measures (e.g. lockdowns) were utilized in analysis. Was it possible to account for the variable duration of such measures, the fact that some of these measures may have been introduced and lifted multiple times over the observation period? And what about the problem of COVID-19 testing capacity, which may have led to underestimate the number of cases in some countries, possibly affecting the decision-making process concerning anti-COVID-19 public health measures? Relatedly, it would be important to provide the definition of each measure of interest (e.g. lockdown, social distancing, etc) in each country, given that between-country differences exist.

7) The authors talk about “notified COVID-19 cases”, which likely represent the number of those testing positive for SARS-CoV-2 regardless of clinical presentation. Unless there is confidence that such notified cases were symptomatic and can thus be labelled as “COVID-19 cases”, I believe that talking about “notified SARS-CoV-2 infections” would be more appropriate.

8) Results, lines 147-149: I would suggest reporting in text the average incidence or number of reported cases in each country. Figure 1 is great and very helpful to see how numbers changed over time, but adding numbers in the description of the results would further clarify the magnitude of the problem.

9) Results, lines 161-168: From this paragraph the reference level is unclear. How did the authors calculate the percent decline at a given timepoint? Was it determined relative to the previous quarter? Or relative to the same quarter of the previous year? Other? I think it is particularly important to clarify this point in part because comparing different quarters would fail to account for the known seasonal variability in TB case detection.

10) Results, lines 225-226: Did these algorithms include the use of integrated testing platforms allowing to detect TB and COVID-19 on the same sample and/or at the same time? Would it be possible to expand a bit on this point and briefly explain how such algorithms were structured? References would also be helpful.

11) Discussion, lines 238-241: I would suggest greater caution in interpreting the study findings regarding the impact of different measures on TB notification levels. To my understanding, various potential confounders, including seasonality, were not accounted for in the models. Before making strong claims on the effect of a given measure, it is important to clearly describe how the impact evaluation was carried out, and – in the event that potentially important factors could not be properly considered – discuss the expected implications for the results. As noted above, perhaps this can be adequately addressed by simply describing the model structure more in detail.

Minor comments:

1) I would suggest checking the paper for minor spelling and grammar errors.

2) In the title, the study country income level should be corrected to “lower-middle income” (instead of “low-middle”), as per World Bank classification system.

3) Line 95: Please add a reference to support the statement that the chosen countries have a high TB burden.

4) Line 139: Does “online dataset” corresponds to “publicly available datasets”? If so, I would opt for the latter in the interest of clarity.

5) I would suggest removing p-values from text and tables. Reporting 95% CIs is more than enough to capture the uncertainty around model estimates, not to mention that a p-value of 0.00 looks quite strange.

6) Discussion, line 250: I would suggest changing to “decline in healthcare seeking” or (better) “change in healthcare seeking behaviours”.

Reviewer #2: Summary

In this study, Spruijt and colleagues described TB case notifications in 4 high TB burden countries, as well as some rates of TB treatment uptake and completion. These trends are graphically illustrated. Data were collected by KNCV country officers. The findings of this study are very interesting and will be valuable to the TB field. However, there are some important shortcomings to the paper, namely with respect to the regression analysis. I don’t think it’s necessarily the right analysis to assess the effect of IPC measures on TB case notifications. Further elaboration is needed in certain areas. Please see below for specific comments.

Specific comments

Title:

Perhaps consider altering title to be “four countries with high TB burdens” instead of referring to income level? Just a thought

Abstract:

Background lists 3 countries but the title suggests that 4 countries are included. I think you forgot Vietnam.

In Results, I would suggest reporting the notification trends in the order that the countries are named in the background. As it stands, the results jump around a bit and only 3 countries are talked about, which left me wondering about Tanzania’s trend.

Also in the Results, it says that “Countries reported the following strategies to retain TB prevention and care services: …” but it is not stated whether these interventions were associated with a change in notifications. Then, the Abstract conclusion states that “Digital health solutions and the integration of COVID-19 and TB testing services have great potential to recover and retain TB care and prevention services”, but this has not been substantiated earlier. Please revise for consistency and completeness.

Introduction:

Is some of the material cited reference 7 more appropriately attributable to the primary source the PDF refers to, i.e., https://www.who.int/docs/default-source/hq-tuberculosis/covid-19-tb-clinical-management-info-note-dec-update-2020.pdf?sfvrsn=554b68a7_0#:%7E:text=TB%20patients%20should%20take%20precautions,cough%2C%20fever%20and%20difficulty%20breathing. ? would suggest including this brief as a citation too

Methods

Line 114-115: the period stated here is Q1 2018 – Q1 2020, but do you mean Q1 2021?

Line 115-116: what is the “Global Tuberculosis”?

Line 115-117: Was it not possible to get more granular data from Viet Nam? Assuming a constant rate (i.e., the annual total divided by 4) of TB notifications throughout 2020 seems like an important assumption that could impact the later analyses and erase some important trends. Any comment on this?

Line 118-121: what countries are we talking about here? Please name in-text

Line 124-125: how did you visualize this? i.e.,what software did you use?

Line 128: what is an ETA coefficient? Some explanation of this measure would be welcome

Line 333: reference 10 needs more detail, I think. I wanted to check what the Pearson scale was but there’s no reference to chapter or page number here. Is this the same as ‘Pearson’s r’?

I’m a bit unclear about the analysis; specifically, why the ETA was done and why Pearson’s correlation scale was used for interpretation of the prior correlation. I think that may be the result of me not understanding the structure of the data, but could you elaborate a bit? Wasn’t the negative binomial regression also run to assess the association between COVID-19 IPC measures and TB notifications?

Am I correct in saying that what we’re trying to understand here is whether IPC measures had an impact, i.e., a causal effect, on TB notifications within particular countries? The IPC measures were policy interventions applied at a group level. I am not an expert in social epidemiology, but to me it seems that the situation at hand, i.e., repeated within-state measures taken before and after well-defined policy changes, could be appropriately analysed using quasi-experimental modeling. I would strongly recommend consulting with a biostatistician, but I think a fixed effects model (i.e., where each country serves as its own control before IPC measures were implemented and case after IPC measures were implemented, therein controlling confounding to a certain degree) may be appropriate. The textbook Methods in Social Epidemiology 2nd Edition (Oaks and Kaufman) has a good chapter on fixed effects and difference in differences models that will likely be informative, https://pubmed.ncbi.nlm.nih.gov/24366487/ may be as well.

Results

Line 147-151: since there are only 4 countries under consideration, for clarity I think it would be better if you just name the countries at issue instead of saying “all countries except for YYY”, since that really only means 2 or 3 countries.

Figure 1: Figure 1 contains a lot of information and is still relatively easy to follow. The main suggestion I have would be to try to get the boxes to be easier to discern from each other. Sometimes it is hard to know what box corresponds to each number. Perhaps the box outlines could each be a different colour and the corresponding number could be written in that colour too? As well, I’d suggest changing the numbers for each IPC to be a letter (e.g., a b c d), since there are already so many numbers in the plot

Table 2: should this be labeled Table 2 when it’s the first table in the paper? Is a table missing earlier? Otherwise, content is quite helpful. Please spell out acronyms in table legend.

Line 171-177: so a separate model was run for each IPC measure? That was not very clear from the methods, please update and specify this.

Table 5 (line 180): why is this table called Table 5? Why is there another Table 5 on line 232? Please check manuscript over for style consistency! Why is it informative to consider these results in terms of the ETA coefficient instead of just interpreting the IRRs? What does the “1” in every cell in the IRR (95% CI) column mean?

Line 186: what does “operate at a lower level” mean? To me, level sounds like “level of the healthcare system” i.e., central or peripheral etc. Low capacity? Please clarify in-text.

Line 186-198: This is all very interesting but it’d not exactly clear to me from the methods how this information was collected. Is this just the “landscape” information gathered by KNCV staff? More details regarding data collection are needed. Was it collected via survey? Administrative data? Interviews? Semi-structured interviews? Some other means? Please elaborate in the methods.

Table 4: what do the dark squares indicate? Suggest instead writing ‘yes’ or ‘no’

Lines 209-230: again, this is really interesting information but it isn’t clear how it was collected and it is not clear if these interventions were implemented across the country or just in select settings. More details are needed to understand if these strategies were deployed throughout countries. This will help readers understand the generalizability of the findings

Other Table 5: what do the dark squares indicate? Suggest instead writing ‘yes’ or ‘no’. Acronyms need to be spelled out in the table legend or as a footnote

Discussion

Lines 235-243: in the results the findings are termed with respect to “association between IPC and change in TB notification’, but in the discussion the findings are termed as “the effect of IPC measures”. This is an important and substantial difference in statistical terminology. The former may well be correct, but I really don’t think with the modeling that’s been done in this paper we can talk at all about “the effect of IPC on the outcome” – I don’t think the appropriate models have been run to talk about causal effect of IPC on TB case notifications. Additionally, no analysis was run that showed that the digital treatment support activities had any effect on TB notifications. The timing of digital interventions may have temporally coincided with rising TB case notifications, but that certainly does not mean that their implementation was the cause behind the notification rise.

Line 245-256: these conclusions are much more reflective of what was actually observed in the study

Line 262-265: again, while I agree that the countries did implement these digital etc interventions, and that the aim was to recover and maintain TB prevention and care services, I don’t think that this is what the paper has shown.

Line 270-272: yes agree that this is what the study showed

Line 274-275: I think that that is fine!

Line 275-287: I agree, the visual depictions are very illustrative. It’s better to not stretch data to reach conclusions that the analyses don’t necessarily support, and to be conservative in conclusions drawn. The descriptive data provided here is informative in and of itself and more attention and space should be given to how exactly it was collected, and those findings could be further elaborated upon. Agree with the conclusions stated here.

How do your results relate to the rest of the literature and state of the field? What are the next steps?

Reviewer #3: Summary:

This study looks at the relation between population-level COVID control measures and TB case notifications to the national TB programs in four high-TB-incidence countries from Q1 2018-Q1 2021, as well qualitative changes in those TB prevention and care programs during the COVID pandemic. Trends were visualized for all 4 countries. A statistical analysis for correlation between COVID interventions and the number of TB notifications during the intervention period was conducted for 2/4 countries. Changes TB service levels as well as adaptations made by TB programs during the study period are described.

The four study countries all had different experiences during the study period. A wide range of COVID interventions (from none to comprehensive) were documented. Two countries experiences a marked drop in TB notifications during the COVID pandemic; the third had fluctuating notifications and the fourth continued the slow increase observed prior to the COVID pandemic. The statistical analysis found no clear relationship between COVID measures and TB case notifications (strong correlation but in opposite directions for the two countries studied). All countries experienced at least some reductions in TB service levels during the study period, and made extensive adaptations to TB services.

Recommendations:

Language and writing:

Major:

1. A thorough copy-edit would help to improve readability and flow – there are many minor grammatical and word choice issues, and it is sometimes hard to follow the sequence of information. As an example, line 38-40 in the abstract reads 'we conducted a country-specific landscape assessment on COVID-19 measures … and employed strategies to recover and retain those services', but I think what the authors mean is '…on COVID-19 measures and strategies employed'.

2. Re flow – it would help the reader follow along to organize the results under sub headings of Descriptive data (Figure 1, Table 2, and accompanying text) then Statistical analyses (The paragraph on impact of COVID-19 IPC measures on TB notification, and the first Table 5, on line 180) .

3. Careful copy-editing to check number consistency is also essential to make sure the information is clear. There are multiple occasions where the numbers given are not consistent, making it difficult to know which is correct (eg line 192: "human resources were affected in four countries (Kyrgystan, Nigeria, and Viet Nam)"). The tables are also mis-numbered (the tables are not numbered sequentially, and there are two Table 5s).

4. There are also some data placeholders in the text still that need to be filled in (eg in supplement 3, for each of the four countries: "the first COVID-19 patient was notified in the XX database [REF]").

Minor:

5. There are a number of abbreviations (eg "M&E") which may not be familiar to all readers; it would be helpful to use the full name instead.

6. The authors should briefly define "notification" in the methods section; this term is commonly used in the TB field but may not be familiar to the general reader. Similarly I recommend the authors include a link to the website of the Global TB Program (as a reference) to aid those unfamiliar with it.

I am not clear what the authors mean by "remote monitor and refill services" inline 222. Does this mean pharmacies delivering medications to the patient's home rather than the patient travelling to the pharmacy?

Methods:

Major:

7. There is a lot of information presented, both qualitative and quantitative; not all of it is directly relevant to the main study objective. While COVID case data provides some context, there is no analysis of COVID case counts in relation to the main topic of TB notifications and TB program functioning. I recommend deleting the COVID case count data from the main manuscript. I think it just distracts from the authors' main focus, especially because COVID case data has all kinds of data completeness issues. It is interesting to include for context in the supplementary material though (supplement 3).

8. Similarly, data on TB treatment initiations and completions are included for 3/4 countries studied, but there is minimal analysis or comment on this aspect. I recommend deleting it from the main text and Figure 1, especially as it's not available for all of the countries. It is interesting to include in the supplemental data though.

Minor:

9. the authors need to clarify the relationship between the "country profiles" in supplement #3 and the "landscape assessment" in supplement #1. I think the narrative, qualitative country profiles were the actual data collected by the KPNV country representative, and then this narrative, qualitative data was extracted into the more structured categorical landscape format to enable tabular and qualitative analysis. However this is not at all clear currently in the methods.

10. The authors should clarify why they used multiple and different sources for TB notification data across the 4 countries and time frame of the study. Why not get quarterly data through the Global TB program for the full study period (2018-2021) for all 4 countries, so there would be more consistency? I can think of potential reasons, but there are none provided currently.

Results:

Major:

11. Figure 1 presents the fundamental quantitative data in this study. The visuals are confusing – I recommend the authors revise the graphics/labelling for the intervention periods (ie when each intervention stopped and started) as this is currently difficult to interpret particularly for Kyrgystan. Using a different coloured horizontal bar (or thick line) for each type of intervention might work better. It is very confusing that some data beyond the study period are shown – eg TB notifications for Tanzania and Viet Nam continue past Q1 2021, and all COVID interventions in all countries appear to end suddenly at Q1 2021 despite other COVID data continuing past this time point. I strongly recommend the authors use a consistent end point for all data throughout the study – presumably this would be Q1 2021.

Minor:

12. Tables 3, 4, and 5 on TB program experiences and strategies are very helpful to organize and visualize this complex set of information. I recommend deleting Table 2, as I found Figure 1 much more useful to portray the same information on COVID intervention timelines in the 4 countries. All the tables should be re-numbered so they are in sequence, including the other "Table 5" on the correlation analysis (there are currently two table 5s)

13. The data on what aspects of TB programs were affected during the pandemic, and strategies used to strengthen services despite these challenges, are the most solid part of this study. This information is also very useful for other TB programs globally. I recommend the authors focus more on these components.

Discussion:

14. The interpretation about correlation between COVID interventions and TB notifications needs to be strengthened. My interpretation is that the overall analysis did not show any clear correlation at all; correlation was strong for both Nigeria and Kyrgystan, but in opposite directions, and wasn't performed for Viet Nam for statistical reasons that I think may not be clear for many readers (defensible statistical choice, but not fully explained in the methods or results). Nevertheless, the question about impact of COVID measures on TB notifications is an urgent one to ask. It would be very useful for the authors to provide their interpretation / speculation on why this analysis was inconclusive/inconsistent.

15. Given that Nigeria was able to maintain gradual increase in TB notifications despite the pandemic, it would be very useful for the authors to include possible reasons why this apparent success happened, in contrast to the experience of the other 3 countries.

6. PLOS authors have the option to publish the peer review history of their article (what does this mean?). If published, this will include your full peer review and any attached files.

Reviewer #1: No

Reviewer #2: No

Reviewer #3: No

---

## [Author Response · Author response to Decision Letter 0]

22 Sep 2022

Manuscript title: Digital health solutions and integrated COVID-19 and TB services to help recover TB care and prevention services in the COVID-19 pandemic: a descriptive study in four high TB burden countries

Manuscript ID: PONE-D-22-12851

September 16, 2022, The Hague, The Netherlands

Dear editor,

Thank you for the opportunity to submit a revised version of our manuscript. We are grateful for the reviewers’ comments, which we believe were helpful to strengthen our paper. We responded to the best of our capabilities. We responded to the comments one-by-one below in the section ‘Response to the Reviewers’. Furthermore, we updated our collected data and consequent analysis to include the periods Q2 2021, Q3 2021, and Q4 2021.

We look forward to your decision regarding acceptance of the revised manuscript. 

Kind regards

Ineke Spruijt

TB consultant and epidemiologist

Postbus 146, 

2501 CC The Hague

Visiting adress: Maanweg 174, 2516 AB, The Hague, The Netherlands

Journal Requirements:

The PLOS ONE style templates can be found at

We have consulted the style templates and have adjusted the manuscript, figures, supplements accordingly.

We have adjusted the Ethics statement. However, we stand by our first explanation that ethical review of anonymous surveillance data and a landscape assessment done by project members (which did not include any personal data) do not require approval of an ethics committee and nor a written or verbal informed consent.

3. In your Data Availability statement, you have not specified where the minimal data set underlying the results described in your manuscript can be found. Upon re-submitting your revised manuscript, please upload your study’s minimal underlying data set as either Supporting Information files or to a stable, public repository and include the relevant URLs, DOIs, or accession numbers within your revised cover letter. 

We have added the minimum data supporting the manuscript in Supporting file 3.

We have included the caption for the supporting information files at the end of the manuscript 

Response to the reviewers:

Reviewer #1: 

I have read with great interest the study by Spruijt and colleagues regarding the impact of COVID-19 on TB services in four high TB burden countries. The research question is of utmost important and the study provides timely findings, showing the importance of implementing digital tools to counteract the negative consequences of the pandemic on TB prevention and care. The article is well-written but would benefit from revision, with particular focus on the methods. In fact, I believe that a more detailed description of the approaches used to address the study objectives would help interpretation and enable replication/reproducibility. 

Thank you for this suggestion. We have re-written the methods section and provided a more detailed description of the different approaches used to study the objectives.

Most of the study is qualitative in nature, aside from the use of statistical modelling to determine associations between various COVID-19-related measures and TB notification rates. The dual nature of the study, which allowed to provide insights into the reasons behind certain events and measures taken to retain TB activities in spite of the ongoing challenges, could be better emphasized as a strength of the study. Thank you, we have highlighted this strength in the discussion section.

Please see more detailed comments/questions below:

Major comments:

1) Abstract: Only three countries are mentioned in the background, but a fourth country (Vietnam) appears in the results, which generates confusion. Also, I think it would be nice to briefly indicate the data sources in the methods, so that readers understand right away whether the study being reported pertains to the national context or specific areas within each country.

We have added Viet Nam to the list of countries in the abstract. Furthermore, we added the following sentence to “Study design and setting of the methods section: “We performed a cross sectional study on the impact of COVID-19 IPC measures on national TB notification trends and assessed COVID-19 and TB integrated approaches in four high TB burden countries: Kyrgyzstan, Nigeria, Tanzania, Viet Nam (1).” [Page X, Lines X-X]

2) Methods, lines 106-108: I wondered whether the authors could elaborate a bit more on the methods used to conduct the landscape assessment. Were TB representatives interviewed or invited to fill out a written questionnaire?

We added the following sentence to the methods section – data collection: “The landscape assessment was conducted through an online questionnaire designed in MS Forms.” [Page X, lines X-X]

3) Methods, lines 124-125: I am not convinced that using the term “trends” is appropriate here, given that – to my understanding – no statistical analyses were undertaken to examine trends (i.e., trend tests or other approaches). I would lean towards using a less specific terminology, e.g., “change over time”.

Thank you for this suggestion. We have adjusted the sentence in the methods section and usage of the term “trends” throughout the manuscript. 

4) Methods, lines 128-130: I think that adding a brief explanation about the reasons for choosing the Eta coefficient would be helpful to the reader (perhaps the non-linear nature of the relationship between variables?). In addition, “correlation” and “association” are not synonyms, so I would not talk about “association” here as the Eta coefficient only reflects correlation.

After of reconsideration and comments of the other reviewers. We have decided to remove the presentation of the ETA coefficient in this manuscript as does not add important information (in addition to the IRR form the Negative Binomial Regression model). 

5) Methods, line 133: While I do understand the need for a negative-binomial model instead of Poisson given the problem of overdispersion, I think the model’s description should be expanded a bit to clarify 1) the purpose of this analysis, 2) the model specification (i.e. which variables were included and why/how these were selected), the estimated measures (crude/adjusted incidence rate ratios and 95% CIs?).

Thank you for these suggestions. We have adjusted the text into the following:

“Third, we assessed the effect of COVID-19 IPC measures being present, including lockdown (yes/no); wearing face masks (yes/no); social distancing (yes/no); limited public transport (yes/no) (independent variables; categorical data) on quarterly reported TB notifications (dependent variable; count data). During exploratory analysis, we detected overdispersion in the data: the variance exceeded the mean and a significant One-Sample Kolmogorov Smirnov Test (Supplement 2). To account for this overdispersion, we employed an univariable negative binomial regression model with log-link functions to estimate crude incidence rate rations with 95% confidence intervals (2).” [Page X, Lines X-X]

6) It remains unclear to me whether and how data on COVID-19 control measures (e.g. lockdowns) were utilized in analysis. Was it possible to account for the variable duration of such measures, the fact that some of these measures may have been introduced and lifted multiple times over the observation period? The COVID-19 IPC measures were included in univariable regression models, in which we assessed the quarterly presence of the IPC measures (dichotomous data – yes/no) with the quarterly change in TB notifications (count data). We added these details to the methods section. [Pages 8, lines 155-181]

And what about the problem of COVID-19 testing capacity, which may have led to underestimate the number of cases in some countries, possibly affecting the decision-making process concerning anti-COVID-19 public health measures? 

We are aware that the actual number of COVID-19 cases may be higher than the SARS-CoV-2 notification data is showing. However, we did not study the association between SARS-CoV-2 infection notifications and installment of IPC measures. But we focused on how the installment of the IPC measures were associated with changes in TB notification numbers. We added a description to the discussion section on the limitation of using SARS-CoV-2 notification data. [Pages 18, lines 356-359]

Relatedly, it would be important to provide the definition of each measure of interest (e.g. lockdown, social distancing, etc) in each country, given that between-country differences exist. We included Table 1 in the manuscript to provide the reader with country-specific definitions of lockdown and social distancing. We added the following sentence to the methods section: “COVID-19 IPC measures consisted of 1) Any type of lockdown; 2) Social distancing; 3) wearing face masks in public spaces; 3) limited availability of public transportation. Table 1 provides a country-specific definition of lockdown and social distancing.” [Methods section, page 4-6, lines 106-117]

7) The authors talk about “notified COVID-19 cases”, which likely represent the number of those testing positive for SARS-CoV-2 regardless of clinical presentation. Unless there is confidence that such notified cases were symptomatic and can thus be labelled as “COVID-19 cases”, I believe that talking about “notified SARS-CoV-2 infections” would be more appropriate.

Thank you for this suggestion. We agree that COVID-19 cases refers to number tested positive. Hence, we adjusted the terminology throughout the manuscript. 

8) Results, lines 147-149: I would suggest reporting in text the average incidence or number of reported cases in each country. Figure 1 is great and very helpful to see how numbers changed over time, but adding numbers in the description of the results would further clarify the magnitude of the problem. Thank you for this suggestion. We have added the numbers representing the increases in SARS-CoV-2 infections to the text [Results section, page 9, lines 196-204]

9) Results, lines 161-168: From this paragraph the reference level is unclear. How did the authors calculate the percent decline at a given timepoint? Was it determined relative to the previous quarter? Or relative to the same quarter of the previous year? Other? I think it is particularly important to clarify this point in part because comparing different quarters would fail to account for the known seasonal variability in TB case detection. We have adjusted the paragraph, and now only reported the highest reported number. [Results section, page 9, lines 196-204]

10) Results, lines 225-226: Did these algorithms include the use of integrated testing platforms allowing to detect TB and COVID-19 on the same sample and/or at the same time? Would it be possible to expand a bit on this point and briefly explain how such algorithms were structured? References would also be helpful. We added the following text to the results section: “In Nigeria and Viet Nam, COVID-19 and TB responses were integrated by adopting an integrated COVID-19/TB screening algorithm for “emergency use approval”. The algorithm is currently under consideration to be included in the ational GeneXpert guideline in Nigeria. In Viet Nam, TB screening was integrated into COVID-19 vaccination sites under the USAID Erase TB project.“ [Results section, page 14, lines 277-281]

11) Discussion, lines 238-241: I would suggest greater caution in interpreting the study findings regarding the impact of different measures on TB notification levels. To my understanding, various potential confounders, including seasonality, were not accounted for in the models. Before making strong claims on the effect of a given measure, it is important to clearly describe how the impact evaluation was carried out, and – in the event that potentially important factors could not be properly considered – discuss the expected implications for the results. As noted above, perhaps this can be adequately addressed by simply describing the model structure more in detail. Thank you for this comment. We have adjusted the methods section and described the model in more detail. Additionally, we adjusted the first paragraph of the discussion section, which now shows “associations” rather than direct impact of COVID-19 IPC measures on TB notifications. 

Minor comments:

1) I would suggest checking the paper for minor spelling and grammar errors. We had the paper proof read by a native speaker.

2) In the title, the study country income level should be corrected to “lower-middle income” (instead of “low-middle”), as per World Bank classification system. We have corrected the study country income level to lower-middle income throughout the document.

3) Line 95: Please add a reference to support the statement that the chosen countries have a high TB burden. We have added the following reference to the manuscript: Global tuberculosis report 2021. Geneva: World Health Organization; 2021.Licence: CC BY-NC-SA 3.0 IGO.

4) Line 139: Does “online dataset” corresponds to “publicly available datasets”? If so, I would opt for the latter in the interest of clarity. We have adjusted “online available data / online dataset to publicly available datasets” throughout the manuscript.

5) I would suggest removing p-values from text and tables. Reporting 95% CIs is more than enough to capture the uncertainty around model estimates, not to mention that a p-value of 0.00 looks quite strange. Although there were no p-values in the text, we removed the p-values from table 5.

6) Discussion, line 250: I would suggest changing to “decline in healthcare seeking” or (better) “change in healthcare seeking behaviours”. We adjusted the text in the manuscript accordingly.

Reviewer #2: Summary

In this study, Spruijt and colleagues described TB case notifications in 4 high TB burden countries, as well as some rates of TB treatment uptake and completion. These trends are graphically illustrated. Data were collected by KNCV country officers. The findings of this study are very interesting and will be valuable to the TB field. However, there are some important shortcomings to the paper, namely with respect to the regression analysis. I don’t think it’s necessarily the right analysis to assess the effect of IPC measures on TB case notifications. Further elaboration is needed in certain areas. Please see below for specific comments.

Specific comments

Title: Perhaps consider altering title to be “four countries with high TB burdens” instead of referring to income level? Just a thought 

Thank you for the suggestion, we have adjusted the title.

Abstract:

Background lists 3 countries but the title suggests that 4 countries are included. I think you forgot Vietnam. Thank you for notifying this, we have adjusted the abstract and included Viet Nam in the list of countries.

In Results, I would suggest reporting the notification trends in the order that the countries are named in the background. As it stands, the results jump around a bit and only 3 countries are talked about, which left me wondering about Tanzania’s trend. We have included Tanzania in the results section of the abstract. We decided to keep he countries grouped according to the change in their TB notification trends. We adjusted the text into the following: “TB notifications declined in Kyrgyzstan and Viet Nam, and (slightly) increased in Nigeria and Tanzania. The changes in TB notifications were associated with the installation of various COVID-19 prevention measures for Kyrgyzstan (declines) and Nigeria (increases).” [Abstract section, Page 2, Lines 44-53]

Also in the Results, it says that “Countries reported the following strategies to retain TB prevention and care services: …” but it is not stated whether these interventions were associated with a change in notifications. We do not have data on when these strategies were implemented. Therefore, we cannot link them to changes in notifications. 

Then, the Abstract conclusion states that “Digital health solutions and the integration of COVID-19 and TB testing services have great potential to recover and retain TB care and prevention services”, but this has not been substantiated earlier. Please revise for consistency and completeness. Thank you for pointing this out. We have adjusted the conclusion of the abstract into the following: “Following the COVID-19 pandemic, we did not observe consistent changes in TB notifications across countries. However, all countries reported lower operating levels of TB prevention and care services. Digital health solutions, community-based interventions, and the integration of COVID-19 and TB testing services were employed to recover and retain those services.” [Abstract section, Pages 2, Lines 54-58]

Introduction:

Is some of the material cited reference 7 more appropriately attributable to the primary source the PDF refers to, i.e., https://www.who.int/docs/default-source/hq-tuberculosis/covid-19-tb-clinical-management-info-note-dec-update-2020.pdf?sfvrsn=554b68a7_0#:%7E:text=TB%20patients%20should%20take%20precautions,cough%2C%20fever%20and%20difficulty%20breathing. ? would suggest including this brief as a citation too. Thank you for this suggestion. We have included the citation in both the Introduction section and Discussion section. 

Methods

Line 114-115: the period stated here is Q1 2018 – Q1 2020, but do you mean Q1 2021? Thank you or detecting this error. We have adjusted the text accordingly.

Line 115-116: what is the “Global Tuberculosis”? Thank you for detecting this error – this should be “Global Tuberculosis Programme”: we adjusted the text accordingly.

Line 115-117: Was it not possible to get more granular data from Viet Nam? Assuming a constant rate (i.e., the annual total divided by 4) of TB notifications throughout 2020 seems like an important assumption that could impact the later analyses and erase some important trends. Any comment on this? No unfortunately, we were not able to obtain quarterly data from the National TB Program of Viet Nam. Therefore, we used TB notification data from WHO. However, WHO only has presented quarterly TB notification data from Q1 2020 onwards on their website. Data prior to Q1 2020 is annual data. Indeed, because we might be missing important trends, we did not include Viet Nam in the statistical analysis. 

Line 118-121: what countries are we talking about here? Please name in-text. We added the text “for all countries” to the sentence.

Line 124-125: how did you visualize this? i.e.,what software did you use? We used Microsoft Excel for this. We adjusted the text accordingly. 

Line 128: what is an ETA coefficient? Some explanation of this measure would be welcome

This is a measure for correlation. However, the ETA coefficient is less informative than the final results of the regression model. Therefore, we have deleted the ETA coefficient from this paper.

Line 333: reference 10 needs more detail, I think. I wanted to check what the Pearson scale was but there’s no reference to chapter or page number here. Is this the same as ‘Pearson’s r’?

I’m a bit unclear about the analysis; specifically, why the ETA was done and why Pearson’s correlation scale was used for interpretation of the prior correlation. 

After of reconsideration and comments of the other reviewers. We have decided to remove the presentation of the ETA coefficient in this manuscript as does not add important information (in addition to the IRR form the Negative Binomial Regression model). 

I think that may be the result of me not understanding the structure of the data, but could you elaborate a bit? Wasn’t the negative binomial regression also run to assess the association between COVID-19 IPC measures and TB notifications? That was indeed the case. We have rewritten the methods section, which now accommodates a clearer description of the type of data collected, the type of analysis in regards to their objectives, and what type of data was used in those analysis. We envision that this adjusted methods section now clearly indicates the structure of the data and the objective of doing the binomial regression. 

Am I correct in saying that what we’re trying to understand here is whether IPC measures had an impact, i.e., a causal effect, on TB notifications within particular countries? The IPC measures were policy interventions applied at a group level. I am not an expert in social epidemiology, but to me it seems that the situation at hand, i.e., repeated within-state measures taken before and after well-defined policy changes, could be appropriately analysed using quasi-experimental modeling. I would strongly recommend consulting with a biostatistician, but I think a fixed effects model (i.e., where each country serves as its own control before IPC measures were implemented and case after IPC measures were implemented, therein controlling confounding to a certain degree) may be appropriate. The textbook Methods in Social Epidemiology 2nd Edition (Oaks and Kaufman) has a good chapter on fixed effects and difference in differences models that will likely be informative, https://pubmed.ncbi.nlm.nih.gov/24366487/ may be as well.

In the statistical analysis of this study, we are assessing whether there is a change in TB notifications that can be associated with the installment of IPC measures. We performed for each country univariable regression analysis (1 per IPC measure). The data that we used as an independent variable is aggregated (quarterly) count data. The appropriate methods for analysis of count data is Poisson regression models or -in case of overdispersion- negative binomial regression models. Our data do not allow for assessing a causal effect, nor do we have individual level data that would allow for fixed effects models. We have adjusted the methods section and incorporated a more detailed description of the data used and type of statistical analysis.

Results

Line 147-151: since there are only 4 countries under consideration, for clarity I think it would be better if you just name the countries at issue instead of saying “all countries except for YYY”, since that really only means 2 or 3 countries. We adjusted the text accordingly throughout the manuscript.

Figure 1: Figure 1 contains a lot of information and is still relatively easy to follow. The main suggestion I have would be to try to get the boxes to be easier to discern from each other. Sometimes it is hard to know what box corresponds to each number. Perhaps the box outlines could each be a different colour and the corresponding number could be written in that colour too? As well, I’d suggest changing the numbers for each IPC to be a letter (e.g., a b c d), since there are already so many numbers in the plot

Thank you for your suggestions to improve readability of Figure 1. We try to avoid using colors to accommodate persons who are colorblind. We changed the numbers of the IPC measures into letters. We also added the quarterly periods in the legend behind the IPC measures.

Table 2: should this be labeled Table 2 when it’s the first table in the paper? Is a table missing earlier? Otherwise, content is quite helpful. Please spell out acronyms in table legend. Apologies for this error, we corrected the numbering of the tables. Furthermore, we have added the acronyms to the table legend.

Line 171-177: so a separate model was run for each IPC measure? That was not very clear from the methods, please update and specify this. Yes, that is indeed what we have done. We have now included a clearer description in the methods section of this analysis. 

Table 5 (line 180): 

why is this table called Table 5? Why is there another Table 5 on line 232? Please check manuscript over for style consistency! Apologies for this error, we corrected the numbering of the tables.

Why is it informative to consider these results in terms of the ETA coefficient instead of just interpreting the IRRs? We calculated the ETA coefficient as pre-analysis to study associations. However, we agree that the ETA coefficient has no added value here and could be removed from the table as the IRR’s provide the information of interest. 

What does the “1” in every cell in the IRR (95% CI) column mean? The one indicates the references category, we have adjusted 1 into REF and included REF in the acronym section. Additionally, we added the 2 categories of the independent variables (yes/no) for clarification.

Line 186: what does “operate at a lower level” mean? To me, level sounds like “level of the healthcare system” i.e., central or peripheral etc. Low capacity? Please clarify in-text. Operational level strategy = the means used by organizations and services to accomplish their overall objectives. We adjusted terminology and added this description to table 3. Furthermore, we adjusted the text into the following: “For Kyrgyzstan, all TB diagnostic and treatment services were reported to operate at a lower operational level (e.g. using less means to accomplish objectives) when compared to pre-pandemic operational levels (Table 3). For Nigeria, Tanzania, and Vietnam, only active TB case finding- and screening services were reported to operate at a lower operational level, whereas other services remained at similar levels when compared to pre-pandemic levels.” [Page 11, Lines: 237-241]

Line 186-198: This is all very interesting but it’d not exactly clear to me from the methods how this information was collected. Is this just the “landscape” information gathered by KNCV staff? More details regarding data collection are needed. Was it collected via survey? Administrative data? Interviews? Semi-structured interviews? Some other means? Please elaborate in the methods. We have now included these descriptions in the methods section of this analysis. [Methods Section; Pages 4-6, lines 115-117]

Table 4: what do the dark squares indicate? Suggest instead writing ‘yes’ or ‘no’. We adjusted the table accordingly.

Lines 209-230: again, this is really interesting information but it isn’t clear how it was collected and it is not clear if these interventions were implemented across the country or just in select settings. More details are needed to understand if these strategies were deployed throughout countries. This will help readers understand the generalizability of the findings The data of these results were collected through the landscape assessment. The objective of this paper is not to be able to generalize results or interventions to other countries. The paper is descriptive in nature and serves as lessons learned and inspiration from the countries included for other countries. in the methods section of this analysis. 

Other Table 5: what do the dark squares indicate? Suggest instead writing ‘yes’ or ‘no’. Acronyms need to be spelled out in the table legend or as a footnote. We adjusted the table accordingly.

Discussion

Lines 235-243: in the results the findings are termed with respect to “association between IPC and change in TB notification’, but in the discussion the findings are termed as “the effect of IPC measures”. This is an important and substantial difference in statistical terminology. The former may well be correct, but I really don’t think with the modeling that’s been done in this paper we can talk at all about “the effect of IPC on the outcome” – I don’t think the appropriate models have been run to talk about causal effect of IPC on TB case notifications. We agree that the paragraph needs reformulating, we assessed an association and are not able to prove any causal effect. Hence, we reformulated the first paragraph in the discussion section. [Pages X, Lines X-X].

Additionally, no analysis was run that showed that the digital treatment support activities had any effect on TB notifications. The timing of digital interventions may have temporally coincided with rising TB case notifications, but that certainly does not mean that their implementation was the cause behind the notification rise. We apologize for this misunderstanding. We did not associated any digital treatment support activities to changes in TB notifications (we only analyzed the association for IPC measures). However, we indicated and discussed that digital support activities could help to retain and recover lower operating TB services in countries. 

Line 245-256: these conclusions are much more reflective of what was actually observed in the study. Thank you. 

Line 262-265: again, while I agree that the countries did implement these digital etc interventions, and that the aim was to recover and maintain TB prevention and care services, I don’t think that this is what the paper has shown. The results of the landscape assessment show that countries followed WHO advice and implemented interventions to retain / recover their TB care and preventions services. 

Line 270-272: yes agree that this is what the study showed. Thank you.

Line 274-275: I think that that is fine! Thank you.

Line 275-287: I agree, the visual depictions are very illustrative. It’s better to not stretch data to reach conclusions that the analyses don’t necessarily support, and to be conservative in conclusions drawn. The descriptive data provided here is informative in and of itself and more attention and space should be given to how exactly it was collected, and those findings could be further elaborated upon. We have re-written the methods section to clarify the type of data collected and analyzed in relation to their objectives.

Agree with the conclusions stated here. Thank you.

Reviewer #3: Summary:

This study looks at the relation between population-level COVID control measures and TB case ndnotifications to the national TB programs in four high-TB-incidence countries from Q1 2018-Q1 2021, as well qualitative changes in those TB prevention and care programs during the COVID pandemic. Trends were visualized for all 4 countries. A statistical analysis for correlation between COVID interventions and the number of TB notifications during the intervention period was conducted for 2/4 countries. Changes TB service levels as well as adaptations made by TB programs during the study period are described.

The four study countries all had different experiences during the study period. A wide range of COVID interventions (from none to comprehensive) were documented. Two countries experiences a marked drop in TB notifications during the COVID pandemic; the third had fluctuating notifications and the fourth continued the slow increase observed prior to the COVID pandemic. The statistical analysis found no clear relationship between COVID measures and TB case notifications (strong correlation but in opposite directions for the two countries studied). All countries experienced at least some reductions in TB service levels during the study period, and made extensive adaptations to TB services.

Recommendations:

Language and writing:

Major:

1. A thorough copy-edit would help to improve readability and flow – there are many minor grammatical and word choice issues, and it is sometimes hard to follow the sequence of information. As an example, line 38-40 in the abstract reads 'we conducted a country-specific landscape assessment on COVID-19 measures … and employed strategies to recover and retain those services', but I think what the authors mean is '…on COVID-19 measures and strategies employed'. We had an editor check the manuscript for inconsistencies.

2. Re flow – it would help the reader follow along to organize the results under subheadings of Descriptive data (Figure 1, Table 2, and accompanying text) then Statistical analyses (The paragraph on impact of COVID-19 IPC measures on TB notification, and the first Table 5, on line 180). Thank you for this suggestion, we have re-organized the results. 

3. Careful copy-editing to check number consistency is also essential to make sure the information is clear. There are multiple occasions where the numbers given are not consistent, making it difficult to know which is correct (eg line 192: "human resources were affected in four countries (Kyrgystan, Nigeria, and Viet Nam)"). The tables are also mis-numbered (the tables are not numbered sequentially, and there are two Table 5s). Apologies for these errors, we have carefully copy-edited the manuscript and corrected the table-numbering.

4. There are also some data placeholders in the text still that need to be filled in (eg in supplement 3, for each of the four countries: "the first COVID-19 patient was notified in the XX database [REF]"). Apologies for these errors, we have carefully copy-edited the manuscript and filled in or removed the placeholders.

Minor:

5. There are a number of abbreviations (eg "M&E") which may not be familiar to all readers; it would be helpful to use the full name instead. We have included description of acronyms in the tables. 

6. The authors should briefly define "notification" in the methods section; this term is commonly used in the TB field but may not be familiar to the general reader. Similarly, I recommend the authors include a link to the website of the Global TB Program (as a reference) to aid those unfamiliar with it. Thank you for this suggestion. We have included an explanation of TB notification data and included a reference to the website of the Global TB Program.

I am not clear what the authors mean by "remote monitor and refill services" in line 222. Does this mean pharmacies delivering medications to the patient's home rather than the patient travelling to the pharmacy? These services can be executed in different ways, but are community-based. We adjusted the sentence into the following: “Furthermore, healthcare facilities in Nigeria limited the need for patients to travel to pharmacies by making more use of community-based treatment monitor and medication refill services.” [Results section, Page 13, Lines 272-274]

Methods:

Major:

7. There is a lot of information presented, both qualitative and quantitative; not all of it is directly relevant to the main study objective. While COVID case data provides some context, there is no analysis of COVID case counts in relation to the main topic of TB notifications and TB program functioning. I recommend deleting the COVID case count data from the main manuscript. I think it just distracts from the authors' main focus, especially because COVID case data has all kinds of data completeness issues. It is interesting to include for context in the supplementary material though (supplement 3).

Thank you for this suggestion. After thorough review of the manuscript and taking into account comments from all three reviewers, we followed your suggestion and excluded the data on COVID from the main manuscript and moved the data collection description to Supplement 3.

8. Similarly, data on TB treatment initiations and completions are included for 3/4 countries studied, but there is minimal analysis or comment on this aspect. I recommend deleting it from the main text and Figure 1, especially as it's not available for all of the countries. It is interesting to include in the supplemental data though. Thank you for this suggestion. After thorough review of the manuscript and taking into account comments from all three reviewers, we followed your suggestion and excluded the TB treatment data from the main manuscript and moved the data collection description to Supplement.

Minor:

9. the authors need to clarify the relationship between the "country profiles" in supplement #3 and the "landscape assessment" in supplement #1. I think the narrative, qualitative country profiles were the actual data collected by the KPNV country representative, and then this narrative, qualitative data was extracted into the more structured categorical landscape format to enable tabular and qualitative analysis. However this is not at all clear currently in the methods. To clarify the supplement 3, we added the following sentence to the methods section: “S3 provides a country profile including detailed results of the changes in SARS-CoV-2 and TB notification data and the landscape assessment.” [Methods section, Page 8, Lines 179-181]

10. The authors should clarify why they used multiple and different sources for TB notification data across the 4 countries and time frame of the study. Why not get quarterly data through the Global TB program for the full study period (2018-2021) for all 4 countries, so there would be more consistency? I can think of potential reasons, but there are none provided currently. We have added the data collection procedures and reasons for collecting data from different sources:

First, we collected data publicly available data from the Global Tuberculosis Program (e.g. period Q1 2020 until Q4 2021 for all countries). Second, as quarterly data was not publicly available through the Global TB Program for the period prior to Q1 2020, we designed a data entry tool in Excel and approached KNCV’s TB representatives to collect those data from the NTPs. For Viet Nam, quarterly data could not be obtained from the NTP for the period Q1 2018-Q1 2020) after which we collected publicly available annual data from the Global Tuberculosis Program and calculated average quarterly numbers by dividing annual TB notification numbers by four [Methods section, Page 7, Lines 132-139]

Results:

Major:

11. Figure 1 presents the fundamental quantitative data in this study. The visuals are confusing – I recommend the authors revise the graphics/labelling for the intervention periods (ie when each intervention stopped and started) as this is currently difficult to interpret particularly for Kyrgystan. Using a different coloured horizontal bar (or thick line) for each type of intervention might work better. 

Thank you for your suggestions to improve readability of Figure 1. We try to avoid using colors to accommodate persons who are colorblind. We changed the numbers of the IPC measures into letters. We also added the quarterly periods in the legend behind the IPC measures.

It is very confusing that some data beyond the study period are shown – eg TB notifications for Tanzania and Viet Nam continue past Q1 2021, and all COVID interventions in all countries appear to end suddenly at Q1 2021 despite other COVID data continuing past this time point. I strongly recommend the authors use a consistent end point for all data throughout the study – presumably this would be Q1 2021. We have updated the data until Q4 2021.

Minor:

12. Tables 3, 4, and 5 on TB program experiences and strategies are very helpful to organize and visualize this complex set of information. I recommend deleting Table 2, as I found Figure 1 much more useful to portray the same information on COVID intervention timelines in the 4 countries. All the tables should be re-numbered so they are in sequence, including the other "Table 5" on the correlation analysis (there are currently two table 5s) Thank you for your suggestion. We removed table 2 (table 1 after renumbering) from the manuscript. We apologize for the errors in numbering of the tables. We renumbered the tables accordingly.

13. The data on what aspects of TB programs were affected during the pandemic, and strategies used to strengthen services despite these challenges, are the most solid part of this study. This information is also very useful for other TB programs globally. I recommend the authors focus more on these components. Thank you for your suggestion. We have restructured the results section and envision this puts more focus to the topic of retaining and recovering TB care and prevention services. 

Discussion:

14. The interpretation about correlation between COVID interventions and TB notifications needs to be strengthened. My interpretation is that the overall analysis did not show any clear correlation at all; correlation was strong for both Nigeria and Kyrgystan, but in opposite directions, and wasn't performed for Viet Nam for statistical reasons that I think may not be clear for many readers (defensible statistical choice, but not fully explained in the methods or results). Nevertheless, the question about impact of COVID measures on TB notifications is an urgent one to ask. It would be very useful for the authors to provide their interpretation / speculation on why this analysis was inconclusive/inconsistent. Thank you for this suggestion. We have added the following sentences to the discussion section: “To assess the impact of COVID-19 IPC measures on TB notifications, future research is needed that take into account those factors and their magnitudes. Furthermore, future research should explore and take into account any programmatic and project interventions aimed at increasing TB notifications that were implemented simultaneously with the COVID-19 IPC measures.” [Discussion section, page 18, lines 362-367]

15. Given that Nigeria was able to maintain gradual increase in TB notifications despite the pandemic, it would be very useful for the authors to include possible reasons why this apparent success happened, in contrast to the experience of the other 3 countries.

We added the following paragraph to the discussion section: “Our study showed increases in TB notifications for Nigeria. This is not in line with a study by Odume et al. (2020), who evaluated two active case finding interventions and showed a decrease in clinic attendance, presumptive TB identification, TB cases detected and TB treatment initiation during the first quarter of the COVID-19 pandemic. These findings were supported by Adwole (2020), who also showed reductions of 35% in presumptive TB and 34% in active TB cases detected in a health facility during the same period. News reports confirm the increase in TB notifications, which were contributed to programmatic and facility-based interventions and integration of TB and COVID-19 measures Lacking scientific evidence, it is unclear to what extend new and additional efforts may have outweighed or masked the effects of COVID-19 on the TB epidemiology in Nigeria.” [Discussion section, page 17, lines 328-336]

---

## [Decision Letter · Decision Letter 1]

19 Dec 2022

PONE-D-22-12851R1Digital health solutions and integrated COVID-19 and TB services to help recover TB care and prevention services in the COVID-19 pandemic: a descriptive study in four high TB burden countriesPLOS ONE

Dear Dr. Spruijt,

Thank you for submitting your manuscript to PLOS ONE. After careful consideration, we feel that it has merit but does not fully meet PLOS ONE’s publication criteria as it currently stands. Therefore, we invite you to submit a revised version of the manuscript that addresses the points raised during the review process. Please see the reviewer's comments below and respond by the editing the manuscript with point by point changes. 

We look forward to receiving your revised manuscript.

Kind regards,

Ankit Jain, M.D.

Academic Editor

PLOS ONE

Journal Requirements:

Reviewers' comments:

Reviewer's Responses to Questions

**Comments to the Author**

1. If the authors have adequately addressed your comments raised in a previous round of review and you feel that this manuscript is now acceptable for publication, you may indicate that here to bypass the “Comments to the Author” section, enter your conflict of interest statement in the “Confidential to Editor” section, and submit your "Accept" recommendation.

Reviewer #1: (No Response)

Reviewer #2: All comments have been addressed

Reviewer #3: All comments have been addressed

Reviewer #4: All comments have been addressed

Reviewer #5: All comments have been addressed

Reviewer #6: All comments have been addressed

2. Is the manuscript technically sound, and do the data support the conclusions?

Reviewer #1: Partly

Reviewer #2: Yes

Reviewer #3: Yes

Reviewer #4: Yes

Reviewer #5: Yes

Reviewer #6: Yes

3. Has the statistical analysis been performed appropriately and rigorously? 

Reviewer #1: No

Reviewer #2: I Don't Know

Reviewer #3: Yes

Reviewer #4: Yes

Reviewer #5: I Don't Know

Reviewer #6: Yes

4. Have the authors made all data underlying the findings in their manuscript fully available?

Reviewer #1: Yes

Reviewer #2: Yes

Reviewer #3: Yes

Reviewer #4: Yes

Reviewer #5: Yes

Reviewer #6: Yes

5. Is the manuscript presented in an intelligible fashion and written in standard English?

Reviewer #1: No

Reviewer #2: Yes

Reviewer #3: Yes

Reviewer #4: Yes

Reviewer #5: Yes

Reviewer #6: Yes

6. Review Comments to the Author

Reviewer #1: I think the manuscript has improved compared to the previous version, but there are points that still need clarification/revision as detailed below.

1. Line 18: perhaps this should read “declined by 18%”?

2. Line 69: what is transmitted is the pathogen, not the disease – please amend.

3. Lines 104-105: I suppose that “wearing face masks in public spaces” refers to whether a mandatory masking policy was in place. If so, I would suggest rephrasing for the sake of clarity.

4. Line 105: please provide a definition of “limited availability of public transportation”. I think this is too vague as stated.

5. Lines 107: what are “MS forms”? Please provide the full name of the platform along with its developer.

6. Lines 99-108: from this paragraph it remains unclear to me whether the questionnaire was filled out by country project officers or by others who were contacted by project officers as part of the landscape assessment.

7. Lines 118-119 and subsequent text: please make sure that the WHO’s Global TB Programme is correctly mentioned (it shouldn’t be “Global TB Program”).

8. Line 136: it is unclear to me why only univariable models were used to evaluate the association between a range of IPC measures and TB notifications. Given that more than one measure was typically implemented in a given country, I would expect the effect of these concomitant measures to sum up, thus becoming hard to disentangle. I believe that the resulting crude associations overestimate the impact of individual approaches. Please explain the rationale for doing so and consider undertaking multivariable analyses as well.

9. Lines 145-146: I am aware that the definition of “overdispersion” is that the “variance exceeds the mean”. However, this could be made clearer in the text, especially for readers who are less familiar with these concepts. For instance, I would suggest clarifying the variance and mean of which variable(s) you are referring to. A bit more of elaboration would help.

10. Lines 167-169: this sentence is confusing – please rephrase. Also, while providing absolute numbers is fine and meaningful, I would add percent increases relative to the number of tests done. Absolute numbers alone may not be fully reflective of the actual epidemic trends. Similar considerations apply to the next few sentences that report findings in the same format.

11. Table 2 is not completely clear. I think it is important to make the term of comparison explicit. This is especially important considering that tables and figures should stand-alone and thus contain all the details needed for interpretation. In this regard, the meaning of “Lower” and “Same” is not straightforward. The table title and notes should be revised accordingly.

12. Please make sure that all acronyms are spelled out at their first appearance in the text (e.g. I am not sure that HCW was defined).

13. Line 212: communicated to whom?

14. Lines 227-229: please clarify what this algorithm entails. Is it based on the use of the GeneXpert platform for the simultaneous detection of TB and SARS-CoV-2 on the same specimen or what?

15. Line 238: “Statistical analysis” looks like an odd heading for a subsection of the results. For consistency, please replace with something that clearly identifies the content of the paragraph as done with previous subsection.

16. Line 239: what does “the installation wearing face masks in public” mean?

17. Line 299: this analysis does not assess the “effect” of X on Y, but the “association” between X and Y. This is a very important point, even more so given that only crude estimates are reported. Please avoid the use of causal language that is clearly not supported by the study design and approaches.

18. There are a few typos and missing words throughout the manuscript. Please review for spelling/grammar.

Reviewer #2: Thanks to the authors for the thorough response to my comments (and those of other reviewers). I think the manuscript's conclusions are now more reflective of the findings and analyses. It's a much smoother manuscript now. The revisions to the methods section help a lot. I am still a bit doubtful about the regression analysis, as I think the qualitative information is more informative and holds more descriptive value. Some further copy-editing would be welcome. Good job

Reviewer #3: the writing is much clearer in the revised version - the manuscript is more focused and flows much better. The methods section in particular is much clearer, and analysis as presented more consistent with the objectives of the study. Fig 1 graphics are also much easier to interpret now. It's a very useful and interesting read!

There are a few small copyedit errors remaining:

Table 1 Kyrgystan, under Lockdown: "to minors unaccompanied" should be "for minors unaccompanied"

Table 1 Viet Nam, under Lockdown: "onmly" should be "only"

line 146: there is an extra period

line 279: "contributed" should be "attributed"

Reviewer #4: This paper has considerable potential to bring this issue into the mainstream through PLOS. I read the study with interest. The authors have already revised and addressed several critical issues raised by other esteemed reviewers. The findings of this study, as revised in Revision -1, meet the standards and would be valuable to the discipline of infectious diseases.

Reviewer #5: Authors have conducted a retrospective study in their manuscript titled "Digital health solutions and integrated COVID-19 and TB services to help recover TB care and prevention services in the COVID-19 pandemic: a descriptive study in four high TB burden countries".The topic is of utmost importance and pandemic did posed unique challenges to identify and treat TB patients thorughout the world. Introduction is well scripted and does intrigue the readers about the importance of topic and the underlying rationale of the study is well justified. Methods are well defined and authors have included most of the of the recommendations of reviewers in their revised manuscripts. Results are detailed and well explained with the help of tables which will be essential for readers to understand. Another way of presenting the results could be dividing the result section based on individual country, but I would leave this consideration to authors discretion. As even in current format they are very intelligibly presented. Limitations are included and are just. Minor spell checking and proof reading is required, especially the spelling of "Vietnam".

Reviewer #6: (No Response)

7. PLOS authors have the option to publish the peer review history of their article (what does this mean?). If published, this will include your full peer review and any attached files.

Reviewer #1: No

Reviewer #2: No

Reviewer #3: No

Reviewer #4: No

Reviewer #5: No

Reviewer #6: **Yes: **Jatinder Singh

---

## [Author Response · Author response to Decision Letter 1]

24 Apr 2023

Manuscript title: Digital health solutions and integrated COVID-19 and TB services to help recover TB care and prevention services in the COVID-19 pandemic: a descriptive study in four high TB burden countries

Manuscript ID: PONE-D-22-12851

April 13, 2023, The Hague, The Netherlands

Dear editor,

Thank you for the opportunity to submit a revised version of our manuscript. We are grateful for the reviewers’ comments, which we believe were helpful to strengthen our paper. We responded to the best of our capabilities. We responded to the comments one-by-one below in the section ‘Response to the Reviewers’. Following the journal requirements, we have updated the following references to meet the Vancouver style. We noticed that reference number 1 and 9 were the same (REF: Global Tuberculosis Report 2021. Geneva: World Health Organization; 2021:Licence: CC BY-NC-SA 3.0 IGO.). We removed number 9 and ensured that number 1 was inserted instead.

Furthermore, we updated our collected data and consequent analysis to include the periods Q2 2021, Q3 2021, and Q4 2021. We were able to retrieve quarterly TB notification data for Vietnam, which enabled us to include Vietnam in the statistical analysis. 

We look forward to your decision regarding acceptance of the revised manuscript. 

Kind regards

Ineke Spruijt

TB consultant and epidemiologist

Postbus 146, 

2501 CC The Hague

Visiting adress: Maanweg 174, 2516 AB, The Hague, The Netherlands

Response to the reviewers

Reviewer #1: I think the manuscript has improved compared to the previous version, but there are points that still need clarification/revision as detailed below.

1. Line 18: perhaps this should read “declined by 18%”? Corrected

2. Line 69: what is transmitted is the pathogen, not the disease – please amend. Amended

3. Lines 104-105: I suppose that “wearing face masks in public spaces” refers to whether a mandatory masking policy was in place. If so, I would suggest rephrasing for the sake of clarity. Amended

4. Line 105: please provide a definition of “limited availability of public transportation”. I think this is too vague as stated. We added: (i.e. reduced availability of public transportation compared to pre-COVID pandemic)

5. Lines 107: what are “MS forms”? Please provide the full name of the platform along with its developer. Adjusted into “Microsoft Forms”.

6. Lines 99-108: from this paragraph it remains unclear to me whether the questionnaire was filled out by country project officers or by others who were contacted by project officers as part of the landscape assessment. Adjusted into: The landscape assessment consisted of an online questionnaire filled in by the country representatives which we designed in Microsoft Forms.

7. Lines 118-119 and subsequent text: please make sure that the WHO’s Global TB Programme is correctly mentioned (it shouldn’t be “Global TB Program”). Adjusted into: “the WHO’s Global TB Programme”

8. Line 136: it is unclear to me why only univariable models were used to evaluate the association between a range of IPC measures and TB notifications. Given that more than one measure was typically implemented in a given country, I would expect the effect of these concomitant measures to sum up, thus becoming hard to disentangle. I believe that the resulting crude associations overestimate the impact of individual approaches. Please explain the rationale for doing so and consider undertaking multivariable analyses as well.

We agree with this comment and should have combined the measures taken during the same quarterly periods (as they yield the same changes in TB notifications during that period and the effect of an individual measure on the notifications cannot be calculated). The measures taken during different quarterly periods should be included in a multivariable model. When adjusting the analysis, we only could employ a multivariable model for Kyrgyzstan. The results of adjusted analysis are presented in the manuscript. [Page 13-14, Lines 253-263 & Table 5] We also adjusted the methods section to accommodate changes made to the analysis. [Page 7-8, Lines 151-161]

9. Lines 145-146: I am aware that the definition of “overdispersion” is that the “variance exceeds the mean”. However, this could be made clearer in the text, especially for readers who are less familiar with these concepts. For instance, I would suggest clarifying the variance and mean of which variable(s) you are referring to. A bit more of elaboration would help.

We have adjusted the following sentence: “During exploratory analysis, we detected overdispersion in the notification data: the variance of the response variable (TB notifications) exceeded that of its mean.” [Page 7, Lines 152-154]

10. Lines 167-169: this sentence is confusing – please rephrase. Also, while providing absolute numbers is fine and meaningful, I would add percent increases relative to the number of tests done. Absolute numbers alone may not be fully reflective of the actual epidemic trends. Similar considerations apply to the next few sentences that report findings in the same format.

Thank you for your suggestion. We adjusted the sentences into the following: “Following the emergence of the COVID-19 pandemic in Q1 2020, the number of notified SARS-CoV-2 infections increased most rapidly from 5399 to 41163 infections (662%)) between Q2 2020-Q3 2020 in Kyrgyzstan; from 28759 to 75284 infections (162%) between Q4 2020-Q1 2021 in Nigeria; and from 14449 to 773703 infections (5255% increase) between Q2 2021 and Q3 2021) in Vietnam. . The increase in SARS-CoV-2 infections in Tanzania started after Q2 2021 (from 0 to 25337 infections) (Fig. 1).” [Page 8, Lines 176-184]

11. Table 2 is not completely clear. I think it is important to make the term of comparison explicit. This is especially important considering that tables and figures should stand-alone and thus contain all the details needed for interpretation. In this regard, the meaning of “Lower” and “Same” is not straightforward. The table title and notes should be revised accordingly.

We adjusted the table title into the following: “Table 2. Operational level strategy of tuberculosis diagnostic and treatment services during the pandemic and compared to the pre-COVID-19 pandemic situation per country.” We adjusted the text in the first row of the table into the following: “Operational level strategy1 compared to the pre-COVID-19 pandemic situation per country”

12. Please make sure that all acronyms are spelled out at their first appearance in the text (e.g. I am not sure that HCW was defined). Done

13. Line 212: communicated to whom? Adjusted into “by HCWs to patients”

14. Lines 227-229: please clarify what this algorithm entails. Is it based on the use of the GeneXpert platform for the simultaneous detection of TB and SARS-CoV-2 on the same specimen or what?

We recognize that the words “screening algorithm” may imply that one specimen is used for testing for both TB and COVID-19. However, in Nigeria and Vietnam, it should be seen more as integration of two screening approaches: one for TB (CXR and GeneXpert) and one for COVID-19 (antigen tests). WE have adjusted the text into the following: “In Nigeria and Viet Nam, COVID-19 and TB responses were integrated by adopting an integrated COVID-19/TB screening for “emergency use approval”. The screening approach, which consists of TB screening using chest X-ray and confirmatory Gene Xpert testing and COVID-19 screening using antigen testing,” 

15. Line 238: “Statistical analysis” looks like an odd heading for a subsection of the results. For consistency, please replace with something that clearly identifies the content of the paragraph as done with previous subsection. We agree that “Statistical analysis” is not appropriate in the results section. We have changed the heading of this subsection into: “The association between COVID-19 IPC measures and TB notifications”

16. Line 239: what does “the installation wearing face masks in public” mean? It refers to the installation of policies. We have adjusted the text into: “the installation of policies for mandatory masking in public and social distancing (≥1 meter), and limited availability of public transportation was significantly”

17. Line 299: this analysis does not assess the “effect” of X on Y, but the “association” between X and Y. This is a very important point, even more so given that only crude estimates are reported. Please avoid the use of causal language that is clearly not supported by the study design and approaches. Apologies for the inaccurate use of language. We have adjusted the sentence into the following: “Our study was only able to statistically assess the association between COVID-19 restrictions and changes in TB notifications in two out of four countries.”

18. There are a few typos and missing words throughout the manuscript. Please review for spelling/grammar. Thank you for pointing out the typos and missing words. A native English speaker has reviewed the manuscript and corrected any errors.

Reviewer #2: Thanks to the authors for the thorough response to my comments (and those of other reviewers). I think the manuscript's conclusions are now more reflective of the findings and analyses. It's a much smoother manuscript now. The revisions to the methods section help a lot. I am still a bit doubtful about the regression analysis, as I think the qualitative information is more informative and holds more descriptive value. Some further copy-editing would be welcome. Good job

Thank you for your compliments and efforts to review our manuscript. The manuscript was once more reviewed by a native English speaker to do some final copy-editing.

Reviewer #3: the writing is much clearer in the revised version - the manuscript is more focused and flows much better. The methods section in particular is much clearer, and analysis as presented more consistent with the objectives of the study. Fig 1 graphics are also much easier to interpret now. It's a very useful and interesting read!

Thank you for your compliments and efforts to review our manuscript.

There are a few small copyedit errors remaining:

Table 1 Kyrgystan, under Lockdown: "to minors unaccompanied" should be "for minors unaccompanied"

Table 1 Viet Nam, under Lockdown: "onmly" should be "only"

line 146: there is an extra period

line 279: "contributed" should be "attributed"

Thank you for pointing out the typos and errors. We have adjusted the text accordingly. We also asked a native English speaker to review the manuscript and corrected any further errors.

Reviewer #4: This paper has considerable potential to bring this issue into the mainstream through PLOS. I read the study with interest. The authors have already revised and addressed several critical issues raised by other esteemed reviewers. The findings of this study, as revised in Revision -1, meet the standards and would be valuable to the discipline of infectious diseases.

Thank you for your compliments and efforts to review our manuscript.

Reviewer #5: Authors have conducted a retrospective study in their manuscript titled "Digital health solutions and integrated COVID-19 and TB services to help recover TB care and prevention services in the COVID-19 pandemic: a descriptive study in four high TB burden countries". The topic is of utmost importance and pandemic did posed unique challenges to identify and treat TB patients throughout the world. Introduction is well scripted and does intrigue the readers about the importance of topic and the underlying rationale of the study is well justified. Methods are well defined and authors have included most of the of the recommendations of reviewers in their revised manuscripts. Results are detailed and well explained with the help of tables which will be essential for readers to understand. Another way of presenting the results could be dividing the result section based on individual country, but I would leave this consideration to authors discretion. As even in current format they are very intelligibly presented. Limitations are included and are just. Minor spell checking and proof reading is required, especially the spelling of "Vietnam".

Thank you for your compliments and efforts to review our manuscript. We have presented individual country results in the supplements. We also asked a native English speaker to review the manuscript and corrected any further errors.

Reviewer #6: (No Response)

---

## [Decision Letter · Decision Letter 2]

24 Oct 2023

Digital health solutions and integrated COVID-19 and TB services to help recover TB care and prevention services in the COVID-19 pandemic: a descriptive study in four high TB burden countries

PONE-D-22-12851R2

Dear Dr. Ineke Spruijt,

We’re pleased to inform you that your manuscript has been judged scientifically suitable for publication and will be formally accepted for publication once it meets all outstanding technical requirements.

Kind regards,

Wenping Gong, Ph.D.

Academic Editor

PLOS ONE

Additional Editor Comments (optional):

Reviewers' comments:

Reviewer's Responses to Questions

**Comments to the Author**

1. If the authors have adequately addressed your comments raised in a previous round of review and you feel that this manuscript is now acceptable for publication, you may indicate that here to bypass the “Comments to the Author” section, enter your conflict of interest statement in the “Confidential to Editor” section, and submit your "Accept" recommendation.

Reviewer #1: All comments have been addressed

Reviewer #2: All comments have been addressed

Reviewer #3: All comments have been addressed

Reviewer #6: All comments have been addressed

2. Is the manuscript technically sound, and do the data support the conclusions?

Reviewer #1: Yes

Reviewer #2: Yes

Reviewer #3: (No Response)

Reviewer #6: Yes

3. Has the statistical analysis been performed appropriately and rigorously? 

Reviewer #1: Yes

Reviewer #2: Yes

Reviewer #3: (No Response)

Reviewer #6: I Don't Know

4. Have the authors made all data underlying the findings in their manuscript fully available?

Reviewer #1: Yes

Reviewer #2: Yes

Reviewer #3: (No Response)

Reviewer #6: (No Response)

5. Is the manuscript presented in an intelligible fashion and written in standard English?

Reviewer #1: Yes

Reviewer #2: Yes

Reviewer #3: (No Response)

Reviewer #6: (No Response)

6. Review Comments to the Author

Reviewer #1: Thank you for clarifying and addressing all my previous questions/comments. Congratulations on this interesting manuscript!

Reviewer #2: I have no further comments. The authors' responses to the recent round of comments from Reviewer 1 are very thorough and I do think have further improved the quality of the paper, particularly regarding the beginning of the results. Limitations are well-described also.

Reviewer #3: (No Response)

Reviewer #6: (No Response)

7. PLOS authors have the option to publish the peer review history of their article (what does this mean?). If published, this will include your full peer review and any attached files.

Reviewer #1: No

Reviewer #2: No

Reviewer #3: No

Reviewer #6: No

---

## [Editor Report · Acceptance letter]

7 Nov 2023

PONE-D-22-12851R2 

Digital health solutions and integrated COVID-19 and TB services to help recover TB care and prevention services in the COVID-19 pandemic: a descriptive study in four high TB burden countries 

Dear Dr. Spruijt:

I'm pleased to inform you that your manuscript has been deemed suitable for publication in PLOS ONE. Congratulations! Your manuscript is now with our production department. 

Kind regards, 

on behalf of

Dr. Wenping Gong 

Academic Editor

PLOS ONE